# Brain-inspired replay for continual learning with artificial neural networks

Gido M. van de Ven ⓘ [1,2 ✉], Hava T. Siegelmann ⓘ [3] & Andreas S. Tolias ⓘ [1,4]

Artificial neural networks suffer from catastrophic forgetting. Unlike humans, when these networks are trained on something new, they rapidly forget what was learned before. In the brain, a mechanism thought to be important for protecting memories is the reactivation of neuronal activity patterns representing those memories. In artificial neural networks, such memory replay can be implemented as 'generative replay', which can successfully – and surprisingly efficiently – prevent catastrophic forgetting on toy examples even in a class-incremental learning scenario. However, scaling up generative replay to complicated problems with many tasks or complex inputs is challenging. We propose a new, brain-inspired variant of replay in which internal or hidden representations are replayed that are generated by the network's own, context-modulated feedback connections. Our method achieves state-of-the-art performance on challenging continual learning benchmarks (e.g., class-incremental learning on CIFAR-100) without storing data, and it provides a novel model for replay in the brain.

[1] Center for Neuroscience and Artificial Intelligence, Department of Neuroscience, Baylor College of Medicine, Houston TX 77030, USA. [2] Computational and Biological Learning Lab, Department of Engineering, University of Cambridge, Cambridge CB2 1PZ, UK. [3] College of Computer and Information Sciences, University of Massachusetts Amherst, Amherst, MA 01003, USA. [4] Department of Electrical and Computer Engineering, Rice University, Houston, TX 77251, USA. ✉email: ven@bcm.edu

Current state-of-the-art deep neural networks can be trained to impressive performance on a wide variety of tasks[1]. But when these networks are trained on a new task, previously learned tasks are typically quickly forgotten[2–4]. Importantly, this 'catastrophic forgetting' is not due to limited network capacity, as the same networks can learn many tasks when trained in an interleaved fashion[5]. In the real world, however, training examples are not presented interleaved but appear in sequences with temporal correlations. One solution would be to store previously encountered examples and revisit them when learning something new. Although such 'replay' or 'rehearsal' solves catastrophic forgetting, the scalability of this solution has been questioned as constantly retraining on all previously learned tasks is highly inefficient and the amount of data that would have to be stored becomes unmanageable quickly[6,7]. Yet, in the brain—which clearly has implemented an efficient and scalable algorithm for continual learning—the reactivation of neuronal activity patterns that represent previous experiences is believed to be important for stabilizing new memories[8–11]. Such memory replay is orchestrated by the hippocampus but also observed in the cortex[12,13], and mainly occurs in sharp-wave/ripples during both sleep and awake[14]. Inspired by this, here we revisit the use of replay as a tool for continual learning in artificial neural networks (ANNs).

As alluded to above, a straight-forward way to add replay to an ANN is to use stored data from previously learned tasks and interleave them with the current task's training data[15–17] (Fig. 1a). Relying on stored data is however undesirable for a number of reasons. Firstly, it is a disadvantage from a machine learning perspective as storing data is not always possible in practice (e.g., due to safety or privacy concerns) and it is problematic when scaling up to problems with very many tasks. Secondly, from a neuroscience perspective, if we hope to use replay in ANNs as a model for reactivation in the brain[5], using stored data is unwanted as it is questionable how the brain could directly store data (e.g., all pixels of an image), while empirically it is clear that human memory is not perfect[18]. As alternative to storing data, here we focus on generating the data to be replayed with a learned generative neural network model of past observations[19–21] (Fig. 1b).

Recent evidence indicates that depending on how a continual learning problem is set up, replay might even be unavoidable[21–24]. Typically, continual learning is studied in a task-incremental learning (Task-IL) scenario[24], in which an agent must incrementally learn to perform several distinct tasks. Although this is a natural scenario for many reinforcement learning problems (e.g., incrementally learning to play Atari games[25]), for classification this scenario is often artificial. Imagine an agent that first learns to classify cats and dogs, and then cows and horses. It seems reasonable to expect that this agent should now also be able to distinguish between cats and cows. In the Task-IL scenario, however, the agent is only expected to be able to solve the exact classification tasks it was trained on. Distinguishing between classes from different learning episodes is only required in the class-incremental learning (Class-IL) scenario[24]. Although this difference might seem subtle, it turns out to dramatically affect the difficulty of a continual learning problem: established machine learning algorithms for continual learning fail in the Class-IL scenario even on seemingly simple toy examples. Generative replay (GR) is currently the only method capable of performing well in this scenario without storing data.

An important potential drawback of GR, however, is that scaling it up to more challenging problems has been reported to be problematic[26,27]. As a direct result, Class-IL with more complex inputs (e.g., natural images) remains an open problem in deep learning, as acceptable performance on such problems has so far only been achieved by methods that explicitly store data[15,17]. In addition, from a neuroscience perspective, the reported inability of replay to scale to more realistic problems in a biologically plausible way (i.e., without storing data) is puzzling as it raises the question how replay could underlie memory consolidation in the brain.

Here, we challenge the unscalability of GR. After first confirming the importance of replay for Class-IL, we report experiments on the MNIST dataset highlighting the surprising efficiency and robustness of replay: replaying just a few or low-quality samples can already be enough. Yet, despite these promising experiments with hand-written digits, we also find that scaling up GR to more complicated problems is not straightforward. To address this, we propose a new variant of GR in which internal or hidden representations are replayed that are generated by the network's own, context-modulated feedback connections. We demonstrate that this brain-inspired replay method achieves state-of-the-art performance on challenging continual learning benchmarks with many tasks (≥100) or complex inputs (natural images) without the need to store data.

## Results

**Comparing continual learning methods.** Our first goal was to compare the performance of GR with that of established continual learning methods. For this, as for the remainder of this study, we focused on image classification based continual learning problems. To quantify performance, we used the average test accuracy over all tasks or classes seen so far. For a justification of this measure and a more detailed discussion of the scope of this study, we refer to the discussion.

For our implementation of GR, we followed the general framework proposed by Shin et al.[20]: besides the main model for solving the tasks (i.e., a classifier), a separate generative model was trained to generate the data to be replayed (Fig. 2). We used a standard variational autoencoder (VAE)[28] as generator (see

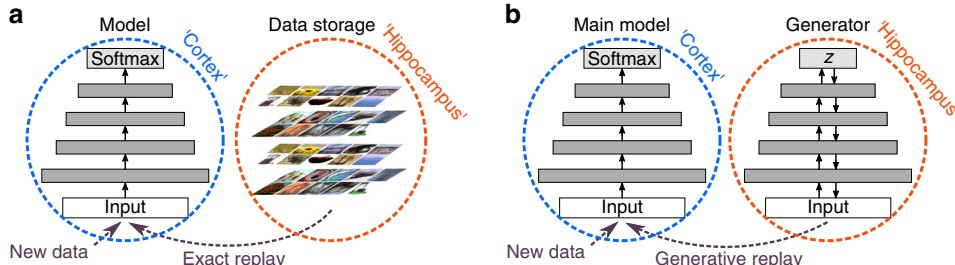

**Fig. 1 Schematic of how current approaches of adding replay to an artificial neural network could be to mapped onto the brain. a** Exact or experience replay, which views the hippocampus as a memory buffer in which experiences can simply be stored, akin to traditional views of episodic memory[77,78]. **b** Generative replay with a separate generative model, which views the hippocampus as a generative neural network and replay as a generative process[62,79].

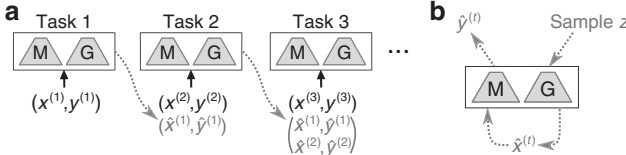

**Fig. 2 Protocol for training artificial neural networks with generative replay. a** On the first task, the main model [M] and a separate generator [G] are trained normally. When moving on to a new task, these two trained models first produce the samples to be replayed (see (**b**)). Those generated samples are then replayed along with the training data of the current task, and both models are trained further on this extended dataset. **b** To produce the samples to be replayed, inputs representative of the previous tasks are first sampled from the trained generator and then labelled based on the predictions made for them by the trained main model.

"Methods"). An alternative strategy for alleviating catastrophic forgetting in ANNs is to protect the parameters of a network that are important for previously learned tasks. Two widely used examples of this regularization-based approach are elastic weight consolidation (EWC)[25] and synaptic intelligence (SI)[29]. Both methods maintain estimates for all parameters of how influential they were for the performance of previous tasks, and those estimates are then used to penalize changes to the more influential parameters when learning a new task. From a neuroscience point of view, these methods can be interpreted as metaplasticity-inspired[30]. Another recent, neuroscience-inspired method for continual learning with ANNs is context-dependent gating (XdG)[31]. To reduce interference between tasks, this method gates for each task a different, randomly selected subset of network nodes. An important limitation of XdG is that it assumes that the specific task to be performed is always known, which means this method cannot be used for Class-IL. The final continual learning method that we considered is learning without forgetting (LwF)[32]. This method has an interesting link with replay-based methods: instead of storing or generating the data to be replayed, this method replays the inputs of the current task after labelling them using the model trained on the previous tasks.

**Class-incremental learning might require replay.** To compare these continual learning methods, we first used the popular deep learning example of classifying MNIST digits[33]. When trained on all digits simultaneously, this is a very simple problem for modern deep neural networks and they make almost no mistakes. But when the dataset is split up into multiple tasks or episodes that must be learned in sequence, the problem becomes substantially more difficult. This task protocol is known as split MNIST[29] (Fig. 3a). Although in recent years it has become a popular continual learning benchmark, it is not always appreciated that this protocol can be setup in multiple ways [or according to different 'scenarios'[24]]. One option is that the network only needs to learn to solve each individual task, meaning that at test time it is always clear from which task the digit to be classified is (i.e., the choice is always just between two possible digits). This is the Task-IL scenario[24] or the 'multi-headed' setup[22]. Another, arguably more realistic option is that the network eventually must learn to distinguish between all ten digits. Set up this way, split MNIST becomes a Class-IL problem. The network must learn a 10-way classifier, but only observes two classes at a time. This more challenging scenario is also referred to as the 'single-headed' setup[22].

On these two scenarios of split MNIST we compared our implementation of GR with EWC, SI and LwF, and on the Task-IL scenario also with XdG. As baselines, we included the naive approach of simply fine-tuning the neural network on each new

task in the standard way (None) and a network that was always trained using the data of all tasks so far (Joint; can be seen as upper bound). For a fair comparison, all methods used similar-sized networks and the same training protocol.

In line with recent reports[21–24], we found that for most of the compared methods there was a dramatic difference in performance between the two scenarios. For the Task-IL scenario, when tasks had to be learned incrementally, all compared methods were successful in preventing catastrophic forgetting (Fig. 3b). Strikingly, however, for the Class-IL scenario, when classes had to be learned incrementally, the metaplasticity-inspired methods EWC and SI dramatically failed and only GR was able to successfully learn all digits (Fig. 3c). This suggests that for Class-IL, when the network must learn to distinguish between classes that are not observed together, some form of replay might be required.

**Efficiency and robustness of generative replay.** These results highlight GR as a promising, perhaps unavoidable, tool for continual learning in ANNs. However, although replaying generated data avoids the issue of having to store potentially large amounts of data, the concern that it is highly inefficient to constantly retrain on all previous tasks is still unaddressed. It is true that for a naive implementation of GR, in which full pseudo-datasets are generated for all previous tasks and concatenated to the current task's training set, this concern certainly applies. Importantly, however, using replay does not necessarily mean fully retraining on all previous tasks. For example, for the implementation of replay used in this paper, in each iteration only a fixed amount of samples were replayed. The total number of replayed samples therefore did not depend on the number of previous tasks. For the results in Fig. 3, every iteration was based on a mini-batch of 128 current samples and 128 replayed samples (divided over the previous tasks). Then, to test whether the amount of replay could be reduced further, we ran additional experiments in which the number of replayed samples per mini-batch was systematically varied. We found that the performance of GR was relatively robust (red lines in Fig. 4a): even with a single replayed sample per mini-batch (i.e., one replayed sample for every 128 samples from the current task), GR performed competitively in the Task-IL scenario and outperformed all non-replay methods in the Class-IL scenario.

Another common criticism of GR is that it simply "shift[s] the catastrophic forgetting problem to the training of the generative model" (ref. [7]; p. 3). The concern here is that training generative models is also a hard problem. That is, although it might be possible to train a model that can generate realistic MNIST images, for real-world problems with more complicated inputs it might be too difficult or computationally be too costly to train high-quality generative models. It is true that the need to train an additional generative model is a disadvantage of GR. But how important is the quality of this generative model? A first indication that replay does not need to be perfect in order to be useful was given by the reasonable performance of LwF on the split MNIST protocol: replaying inputs from the current task—for example replaying '2's and '3's in order not to forget about '0's and '1's—outperformed EWC and SI (Fig. 3b, c). Then, to further and more systematically test how good the quality of the replay needs to be, we varied the number of hidden units in the VAE model used for producing the replay. Strikingly, reducing the VAE's hidden layers to only 10 units resulted in low-quality samples (Fig. 4c; left panel), but only moderately affected the performance of GR (Fig. 4b).

Why is it possible to replay so few or such low-quality examples? One likely reason is that having to learn something new is substantially harder than not forgetting it. This intuition

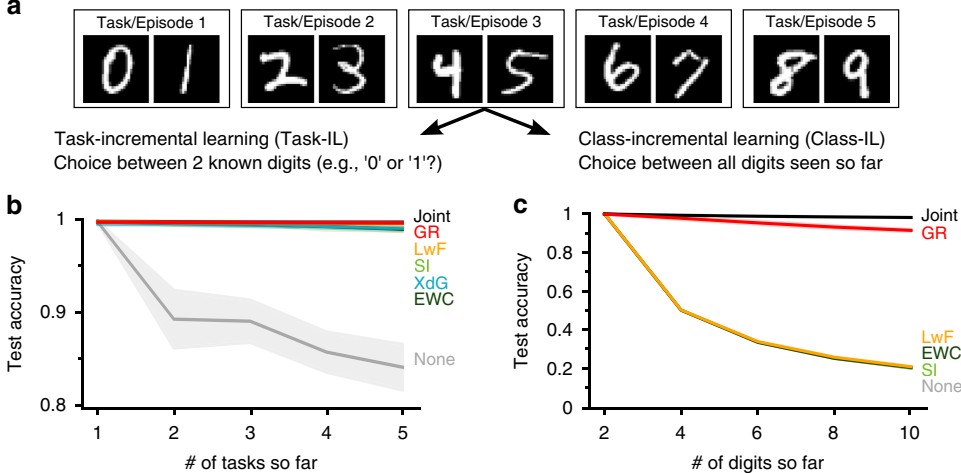

**Fig. 3 Replay might be required for artificial neural networks to incrementally learn new classes. a** The split MNIST task protocol performed according to two different scenarios. **b** In the task-incremental learning scenario (Task-IL), all compared continual learning methods perform very well. **c** In the class-incremental learning scenario (Class-IL), only generative replay (GR) prevents catastrophic forgetting. Reported is the average test accuracy based on all tasks/digits so far. Displayed are the means over 20 repetitions, shaded areas are ±1 SEM. Joint: training using all data so far (`upper bound'), LwF learning without forgetting, SI synaptic intelligence, XdG context-dependent gating, EWC elastic weight consolidation, None: sequential training in standard way.

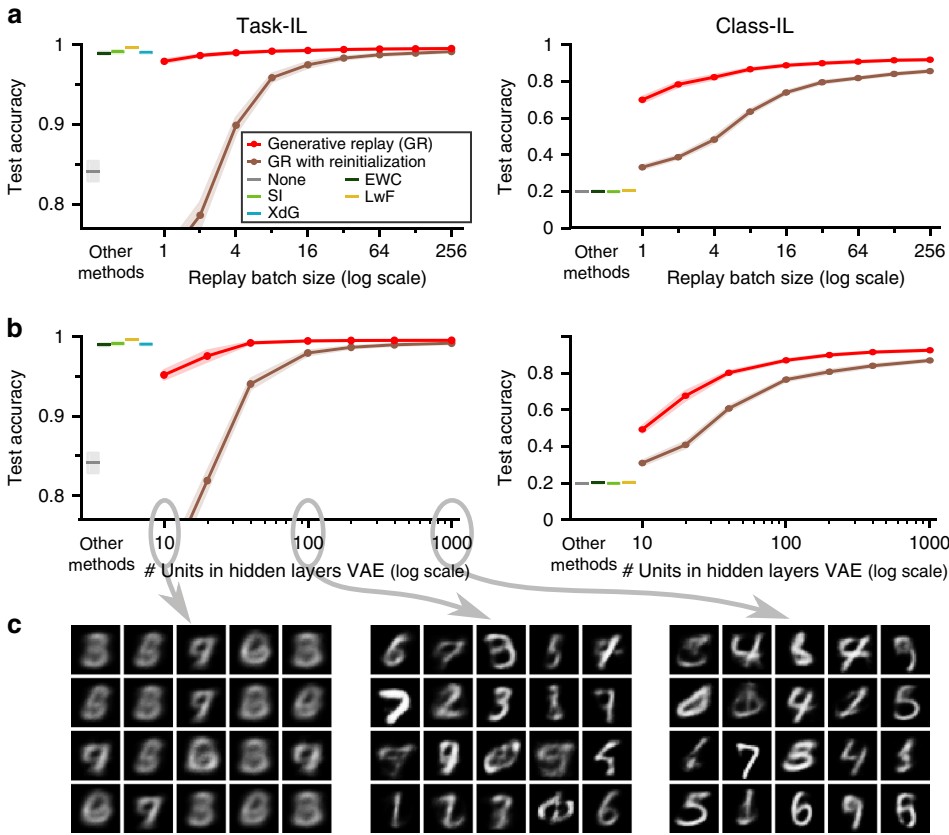

**Fig. 4 Generative replay is remarkably efficient and robust.** It is possible to substantially reduce the quantity or the quality of replay without severely affecting performance. **a**, **b** Shown is the average test accuracy (based on all tasks/digits) of generative replay on the split MNIST protocol performed according to the task-incremental learning scenario (Task-IL; left) and the class-incremetnal learning scenario (Class-IL; right), both **a** as a function of the total number of replayed samples per mini-batch and **b** as a function of the number of units in the hidden layers of the variational autoencoder (VAE) used for generating replay. As a control, also shown is a variant of generative replay whereby the networks are reinitialized before each new task/episode. For comparison, on the left of each graph the average test accuracy of the other methods is indicated (see also Fig. 3). Displayed are the means over 20 repetitions, shaded areas are ±1 SEM. Panel **c** shows random samples from the generative model after finishing training on the fourth task (i.e., examples of what is replayed during training on the final task) for a VAE with 10, 100 and 1000 units per hidden layer, illustrating the low quality of the samples being replayed. None: sequential training in standard way, SI synaptic intelligence, XdG context-dependent gating, EWC elastic weight consolidation, LwF learning without forgetting.

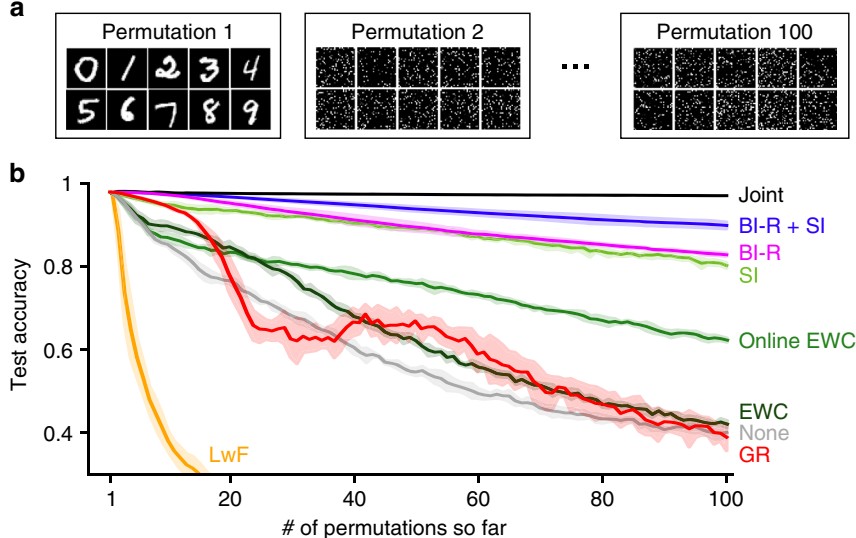

**Fig. 5 Brain-inspired modifications enable generative replay to scale to problems with many tasks. a** The permuted MNIST protocol with 100 permutations. **b** When performed according to the Domain-IL scenario (i.e., no task labels available at test time), the best performing current method is synaptic intelligence (SI). Although standard generative replay (GR) outperforms SI for the first 10 tasks, its performance rapidly degrades after ~15 tasks. With our brain-inspired modifications (BI-R; see below), generative replay outperforms SI also after 100 tasks. Combining BI-R with SI results in a further boost in performance. Learning without forgetting (LwF) performs badly on this task protocol because between tasks the inputs are completely uncorrelated. Reported is average test accuracy based on all permutations so far. Displayed are the means over 5 repetitions, shaded areas are ±1 SEM. Joint: training using all data so far (`upper bound'), EWC elastic weight consolidation, None: sequential training in standard way.

was confirmed by a control experiment in which we re-initialized the parameters of the network before training on each new task (but after the samples to be replayed were generated), so that the network had to constantly re-learn all previous tasks from scratch (brown lines in Fig. 4a, b). In this case, replaying only a few or low-quality examples was indeed not enough to achieve the same strong performance.

**Scaling up to more challenging problems**. To summarize, catastrophic forgetting in ANNs can be prevented by relatively small amounts of 'good enough' replay. This suggests GR has the potential to also be useful for more challenging, real-world continual learning problems. This is what we set out to test next.

We first asked how GR scales to problems with many tasks. A suitable task protocol to test this was permuted MNIST[34] (Fig. 5a), another common benchmark for continual learning. This protocol also uses the MNIST dataset, but now every task contains all 10 digits and for each task a different permutation is applied to the pixels of all images. The goal is always to identify the original digit (i.e., a 10-way classification). Typically, at test time the network is not told which permutation was applied to the image, meaning that the problem is performed according to the domain-incremental learning (Domain-IL) scenario[24]. Although permuted MNIST is usually considered with a limited number of permutations (≤10), a recent study by Masse et al.[31] used this protocol to assess how methods behave when the number of tasks is substantially increased. They found that after 100 tasks, SI and online EWC, a more efficient version of EWC[7], obtained average accuracies of ~82% and ~70%, respectively. Although their proposed method XdG by itself performed worse than both methods (~61%), they found that the performance of SI could be substantially improved (to ~95%) by combining it with XdG. One important caveat, however, is that XdG assumes that the network is always told—also at test time—which permutation was applied to the image. That is, while SI and online EWC by themselves were performed according to the Domain-IL scenario, the variants with XdG were performed according to the easier

Task-IL scenario. Put differently, the gain in performance obtained with XdG was dependent on the availability of additional information at test time. Here, using the same permuted MNIST task protocol with 100 different permutations, we asked whether GR could match or improve upon the performance of SI without using additional information at test time. Unfortunately, despite the promising results presented in the previous section, we found that a straight-forward implementation of GR did not scale well to such a long series of tasks, as its performance rapidly declined after about 15 tasks (red line in Fig. 5b).

A second question is whether GR can scale to problems with more complex inputs. To test this, we used the CIFAR-100 dataset[35] split up into 10 tasks with 10 natural image classes each (Fig. 6a). As with split MNIST, the difficulty of this task protocol differs widely depending on the scenario according to which it is performed. When performed according to the Task-IL scenario (i.e., with the choice always only between 10 classes), we found that methods such as EWC, SI and LwF prevented catastrophic forgetting almost fully (Fig. 6b). The standard version of GR, however, failed on this task protocol with natural images even in the Task-IL scenario. When performed according to the Class-IL scenario (i.e., with the choice between all classes seen so far; so a 100-way classification in the end), the split CIFAR-100 protocol became substantially more challenging and all compared existing methods (i.e., GR, EWC, SI and LwF) performed poorly, suffering from severe catastrophic forgetting (Fig. 6c). Currently, the only methods obtaining acceptable performance on this benchmark are methods that explicitly store data, such as iCaRL[15] or experience replay[17].

**Brain-inspired modifications to GR**. These results, together with other recent reports[26,27], indicate that straight-forward implementations of GR break down for more challenging problems. Although we found earlier that replay does not need to be perfect, it seems that for these problems the quality of the generated inputs that were replayed was just too low (Fig. 6d). One possible

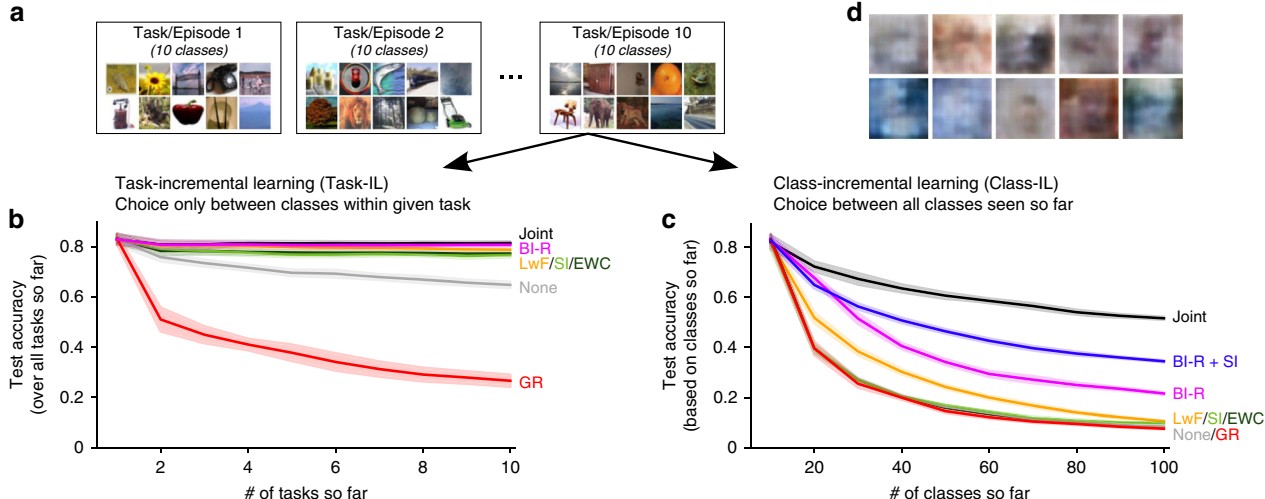

**Fig. 6 Brain-inspired modifications enable generative replay to scale to problems with complex inputs. a** The split CIFAR-100 protocol performed according to two different scenarios. **b** In the task-incremental learning scenario (Task-IL), most continual learning methods are successful, although standard generative replay (GR) performs even worse than the naive baseline. But with our brain-inspired modifications (see below), generative replay outperforms the other methods. **c** In the class-incremental scenario (Class-IL), no existing continual learning method that does not store data is able to prevent catastrophic forgetting. Our brain-inspired replay (BI-R; see below), especially when combined with synaptic intelligence (SI), does achieve reasonable performance on this challenging, unsolved benchmark. Reported are average test accuracies based on all tasks/classes so far. Displayed are the means over 10 repetitions, shaded areas are ±1 SEM. **d** Examples of images replayed with standard generative replay during training on the final task. Joint: training using all data so far (`upper bound'), LwF learning without forgetting, EWC elastic weight consolidation, None: sequential training in standard way.

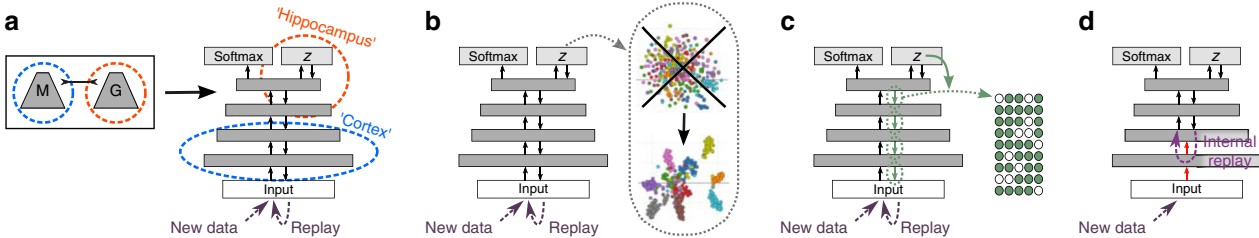

**Fig. 7 The proposed brain-inspired modifications to the standard generative replay framework. a** Replay-through-feedback. The generator is merged into the main model by equipping it with generative feedback or backward connections. **b** Conditional replay. To enable the model to generate specific classes, the standard normal prior is replaced by a Gaussian mixture with a separate mode for each class. **c** Gating based on internal context. For every task or class to be learned, a different subset of neurons in each layer is inhibited during the generative backward pass. **d** Internal replay. Instead of representations at the input level (e.g., pixel level), hidden or internal representations are replayed.

solution would be to try to use the recent progress in generative modelling with deep neural networks[36–38] to improve the quality of the generator. Although this approach might work to certain extent, an issue is that incrementally training high-quality generative models is a very challenging problem as well[26,27]. Moreover, such a solution would not be very efficient, as state-of-the-art generative models tend to be computationally very costly to train or to sample from. Instead, given that the brain has implemented an efficient algorithm for continual learning that is thought to rely on replay, we turned for inspiration to the brain.

Our first modification to the standard GR approach was motivated by anatomy. Replay in the brain originates in the hippocampus, from where it propagates to the cortex[39]. For current versions of GR it has been suggested that the generator, as the source of replay, is reminiscent of the hippocampus and that the main model corresponds to the cortex (Fig. 1b). Although this analogy has some merit, an issue is that it ignores that the hippocampus sits atop of the cortex in the brain's processing hierarchy[40]. Instead, we propose to merge the generator into the main model, by equipping it with generative backward or feedback connections. The first few layers of the resulting model can then be interpreted as corresponding to the early layers of the

visual cortex and the top layers as corresponding to the hippocampus (Fig. 7a). We implemented this 'replay-through-feedback' model as a VAE with added softmax classification layer to the top layer of its encoder (see "Methods").

One issue with a standard VAE is that it is not possible to intentionally generate examples of a particular class. Humans however have control over what memories are recalled[41]. To enable our model to control what classes to generate, we replaced the standard normal prior over the VAE's latent variables by a Gaussian mixture with a separate mode for each class (Fig. 7b; see "Methods"; see also ref. [42]). This made it possible to generate specific classes by restricting the sampling of the latent variables to their corresponding modes. In addition, for our replay-through-feedback (RtF) model, such a multi-modal prior should encourage a better separation of the internal representations of different classes, as they no longer all have to be mapped onto a single continuous distribution.

The brain processes stimuli differently depending on the context or the task that must be performed[43,44]. Moreover, contextual cues (e.g., odours, sounds) have been shown to bias what memories are replayed[45,46]. A simple but effective way to achieve context-dependent processing in an ANN is to fully gate

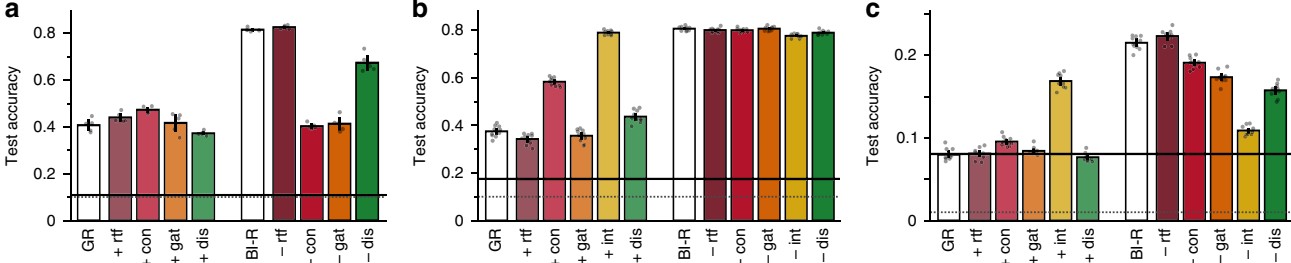

**Fig. 8 Addition- and ablation-experiments to tear apart the contributions of individual modifications.** The average overall accuracy is shown for standard generative replay (GR) with individual modifications added (`+`, left) and for brain-inspired replay (BI-R) with individual modifications removed (`−`, right) for **a** permuted MNIST with 100 permutations, **b** the task-incremental learning scenario on CIFAR-100 and **c** the class-incremental learning scenario on CIFAR-100. Note that internal replay was not used for permuted MNIST, as no convolutional layers were used for this protocol. Each bar reflects the mean over 5 (permuted MNIST) or 10 (CIFAR-100) repetitions, error bars are ± 1 SEM, individual repetitions are indicated by dots. Dotted grey lines indicate chance level. Solid black lines show performance when the base network is trained only on the final task/episode (mean over 5 or 10 repetitions, shaded areas are ±1 SEM), which can be interpreted as chance performance on all but the last seen data. rtf replay-through-feedback, con conditional replay, gat gating based on internal context, int internal replay, dis distillation.

(or 'inhibit') a different, randomly selected subset of neurons in each hidden layer depending on which task should be performed. This is the approach of XdG[31]. But, as discussed above, this technique can only be used when task identity information is always available, which is not the case for domain-IL or class-IL. However, we realized that in these scenarios it is still possible to use context gates—albeit only in the decoder part of our network—by conditioning on an internal context. The internal context that we conditioned on was the specific task or class to be generated or reconstructed (Fig. 7c; see "Methods"). Note that this is possible because for inference (i.e., classifying new inputs) only the feedforward layers of the network are needed, and those layers were not gated.

Our fourth and final brain-inspired modification was to replay representations of previously learned classes not all the way to the input level (e.g., pixel level), but to replay them internally or at the 'hidden level' (Fig. 7d; see "Methods"). Motivation for this was that the brain is also not thought to replay memories all the way down to the input level. Mental images are for example not propagated to the retina[47]. From a machine learning point of view the hope was that generating such internal representations would be substantially easier, since the purpose of the early layers of a neural network is to disentangle the complex input level representations. A likely requirement for this internal replay strategy to work is that there are no or very limited changes to the first few layers that are not being replayed. From a neuroscience perspective this seems realistic, as the representations extracted by the brain's early visual areas are indeed not thought to drastically change in adulthood[48,49]. To simulate development, we pre-trained the convolutional layers of our model on CIFAR-10, a dataset containing similar but non-overlapping images compared to CIFAR-100[35]. During the incremental training on CIFAR-100, those convolutional layers were frozen and we replayed only through the fully connected layers. For a fair comparison, all other methods also used pre-trained convolutional layers for the CIFAR-100 experiments. As no convolutional layers were used with split and permuted MNIST, pre-training and internal replay were not used in those experiments.

Finally, we also made a modification inspired by the machine learning literature. Instead of labelling the generated data as the most likely class according to the main model ('hard targets'), we labelled them with the predicted probabilities for all possible classes ('soft targets'; see "Methods"). This is called 'distillation' and has been shown to be an effective way of transferring information or knowledge from one model to another model[50]. Especially when the quality of the generated data is low, we

expected this way of labelling replayed samples to be important, since it might be harmful to label ambiguous inputs (e.g., that are in between two or more classes) as belonging to a single class.

**Evaluating brain-inspired replay.** To test the effectiveness of these modifications, we applied the resulting brain-inspired replay method on the same benchmarks as before while using similar-sized networks. On the permuted MNIST protocol with 100 permutations, we found that brain-inspired replay out-performed the already strong performance of SI (Fig. 5b). Combining brain-inspired replay with SI pushed performance even higher, achieving state-of-the-art performance on this benchmark when task identity information is not available at test time.

On the split CIFAR-100 protocol, our brain-inspired modifications also substantially improved the performance of GR. In the Task-IL scenario, brain-inspired replay almost fully mitigated catastrophic forgetting and outperformed EWC, SI and LwF (Fig. 6b). In the Class-IL scenario, brain-inspired replay also outperformed the other methods, although its performance still remained substantially under the 'upper bound' of always training on the data of all classes so far (Fig. 6c). Nevertheless, we are not aware of any continual learning method that performs better on this challenging problem without storing data. Finally, as for permuted MNIST, combining brain-inspired replay with SI again substantially improved performance, further closing the gap towards the upper bound of joint training (Fig. 6c).

**Lesion experiments.** To gain insight into the contributions of the various components of our brain-inspired replay method, we performed a series of addition- and ablation-experiments (Fig. 8). Internal replay appeared to be the most influential modification as introducing or removing this component had the largest effects on performance, but we also found that the different modifications were complementary to each other. For both permuted MNIST (Fig. 8a) and the Class-IL scenario on CIFAR-100 (Fig. 8c), the gain in performance obtained by combining all components together was larger than the sum of the effects of adding each of them in isolation. Especially the benefit of conditional replay and that of gating based on internal context depended on their combination with other components. Moreover, for both permuted MNIST and Class-IL on CIFAR-100, none of the individual modifications were sufficient to achieve the performance of brain-inspired replay, while all of them—with the exception of RtF—were necessary. The contribution of RtF is rather that it increases efficiency (i.e., removing the need for a

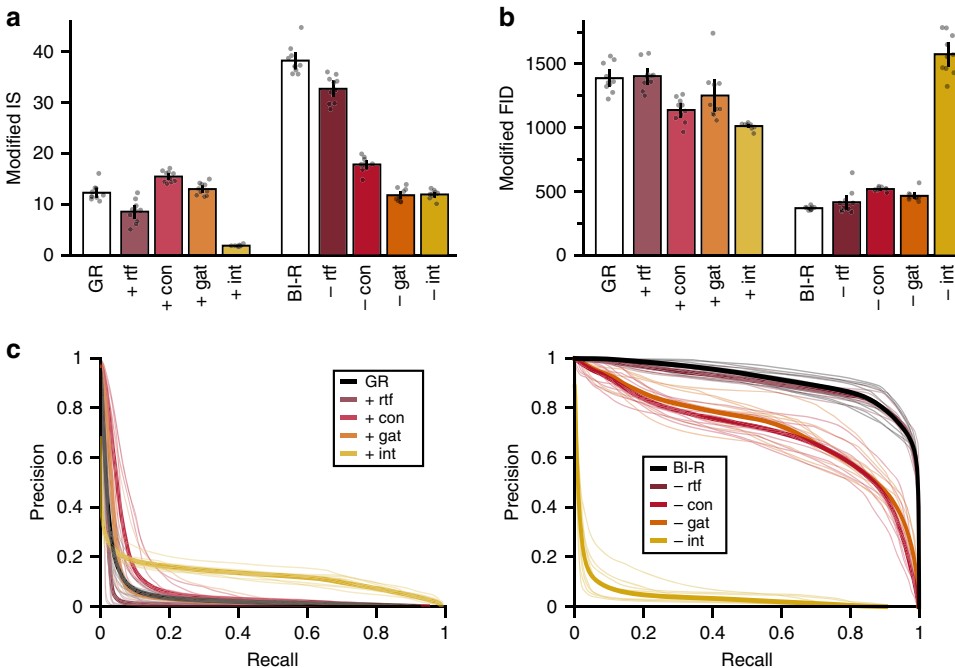

**Fig. 9 Quality and diversity of generated samples.** For the class-incremental learning scenario on CIFAR-100, compared are the quality and diversity of the generated samples that are replayed by standard generative replay (GR) with individual modifications added ('+', left within each panel) and by brain-inspired replay (BI-R) with individual modifications removed ('−', right). All measures are computed for samples generated by the model after it was incrementally trained on all 100 classes; for the measures in the last two panels those generated samples were compared to real samples from the test set. **a** Our modified version of Inception Score (Modified IS). Higher means better samples. **b** Our modified version of Frechet Inception Distance (Modified FID). Lower means better samples. **c** Precision & Recall curves. Towards the top indicates better sample quality (or 'precision'), towards the right indicates better sample diversity (or 'recall'). Each bar or thick curve reflects the mean over 10 repetitions, error bars are ±1 SEM, individual repetitions are indicated by dots or thin curves. rtf replay-through-feedback, con conditional replay, gat gating based on internal context, int internal replay, dis distillation.

separate generative model) without substantially hurting performance. Although when brain-inspired replay was combined with SI, ablating RtF did slightly reduce the average accuracy (permuted MNIST: $0.904 \pm 0.005$ versus $0.892 \pm 0.004$; Class-IL on CIFAR-100: $0.344 \pm 0.002$ versus $0.334 \pm 0.002$), so the best overall performance was in fact obtained with RtF. Finally, that none of the individual components were necessary for the Task-IL scenario on CIFAR-100 (Fig. 8b) reflects that preventing catastrophic forgetting in this scenario is substantially easier.

**Quality of generated replay.** To better understand why brain-inspired replay performed so well, we set out to compare the quality of the generator with and without the various modifications. In particular, we asked whether replaying representations internally rather than at the pixel level improved the quality of the generated samples. Testing for this was however not straightforward. Visually comparing samples was not possible as it is unclear how the generated internal representations could be visualized, while traditional quantitative measures for evaluating VAEs such as reconstruction error or average log-likelihood are problematic for comparing between models of different input distributions (although these measures can be computed for both internal replay and replay at the pixel level, see Supplementary Fig. 4). To fairly compare samples generated at different levels, they first had to be transformed to a common embedding space. Incidentally, the first step for computing recent measures for evaluating generative models such as Inception Score (IS)[51], Fréchet Inception Distance (FID)[52] and Precision & Recall curves[53] is to embed samples into a different feature space. The original versions of these measures embed samples using Inception Net[54], but our generated internal representations cannot be

fed into this network. We therefore replaced the Inception Net with a different neural network with the same pre-trained convolutional layers as the incrementally trained models (see "Methods", see also ref. [55]), so that samples at both the pixel and the internal level could be embedded by this network.

For the Class-IL scenario on CIFAR-100, our modified IS (Fig. 9a) and modified FID (Fig. 9b) measures both indicated that the samples replayed by the brain-inspired replay method were indeed better than those replayed by standard GR. In line with the results in Fig. 8c, this improvement was for a substantial part due to replaying at the internal rather than at the pixel level, although the other modifications (and their combination) also contributed. However, one issue with these two measures is that they only quantify generator performance with a single number; they do not distinguish between sample quality and sample diversity. To test whether the observed improvements in generated samples were mainly driven by better quality or by better diversity, Fig. 9c reports our version of Precision & Recall curves. These curves indicate that our modifications improved both the quality and the diversity of the generated samples to similar extents.

## Discussion
Catastrophic forgetting in ANNs is a major obstacle to the development of artificial agents that can incrementally learn from their experiences[6,56]. Biological neural networks are superior to their artificial counterparts when it comes to continual learning, making it no surprise that the brain has inspired recent attempts to alleviate catastrophic forgetting in ANNs: regularization-based methods such as EWC[25] and SI[29] model the complexity of biological synapses; while the brain's ability to process stimuli differently depending on context inspired explicit task-based

methods such as XdG[31]. We demonstrated that although these methods are successful for scenarios in which tasks must be learned incrementally, they are unable to incrementally learn new classes. Only another neuroscience-inspired approach, replaying examples representative of previous observations, was found to be able to solve Class-IL problems. Critically, such replay does not need to rely on stored data, as it is possible to generate the samples to be replayed. Further strengthening the case for replay, we found that replaying low-quality samples or including just a single replayed example per mini-batch could be sufficient. Nevertheless, it turned out that straight-forward implementations of GR break down for more challenging problems (e.g., with natural images as inputs). To address this, using inspiration from the brain, we proposed a series of simple, easy-to-implement and efficient modifications, and we showed that these enable GR to successfully scale to problems with many tasks or complex inputs.

The continual learning problems considered in this study all involved image classification, raising the question to what extent the insights gained here apply to other machine learning domains or other input modalities. Firstly, although classification was indeed the end goal, we also showed that GR facilitates incrementally learning a generative model and that our brain-inspired modifications improve the quality of the learned generative model. However, it should be noted that in semi- or unsupervised settings, the conditional replay and the gating based on internal context components would need to be modified as their current implementation depends on the availability of class labels during training. It might be possible to instead condition on other context information. In reinforcement learning replaying episodes of previous experiences already is a widely used tool[16,57], but so far it has relied on memory buffers storing these episodes. Future work could test whether brain-inspired GR could remove or reduce this dependence on stored data. Secondly, regarding input modalities, although indeed only image-based experiments were reported in this study, most of the results and proposed methodology should extent to other modalities. In particular, none of the experiments on MNIST used tools specific for images. Only translating the internal replay component to other input modalities will not be straight-forward, as this one depends on pre-trained convolutional layers. Analogous to the separate sensory processing areas in the brain, other input modalities require different pre-processing layers, and it remains to be confirmed whether replaying internal representations will work with those. Another drawback of internal replay is that the rigid, pre-trained convolutional layers likely restrict the ability of the model to learn out-of-distribution inputs (e.g., images without natural image statistics), although to some degree this is true for the brain as well[58]. Finally, another limitation of the current study is that continual learning performance was only quantified by the average accuracy over all tasks or classes seen so far, which is a measure that mainly reflects the extent to which a method suffers from catastrophic forgetting. Other critical aspects of continual learning such as forward and backward transfer or compressability were not explicitly addressed. Especially for Task-IL—where catastrophic forgetting can be prevented by simply training a different network for each task to be learned—it can be argued that these other aspects are most interesting[59]. For Class-IL, however, preventing catastrophic forgetting is still an unsolved problem, justifying our focus on the average accuracy measure.

Besides using insights from neuroscience to improve continual learning with ANNs, another aim of this work was to generate new perspectives and hypotheses about the computational role and possible implementations of replay in the brain. With regard to replay's implementation, this study firstly provides evidence that replay might indeed be a feasible way for the brain to combat catastrophic forgetting. Although it has long been known that revisiting previously seen examples could prevent catastrophic forgetting in toy examples[5], it had remained an open question whether replay could scale to more complex, real-world problems without having to rely on biologically implausible mechanisms such as explicitly storing past observations. Further, our work postulates replay in the brain to be a generative process. This conjecture is in agreement with a growing body of experimental work reporting that the representations replayed in the brain do not directly reflect experiences[60,61], but that they might be samples from a learned model of the world[62–64]. Regarding replay's function, our findings highlight an important computational role for replay in incrementally learning new classes or categories. Being able to distinguish items or objects from each other is critical to survival[65], and there is a vast cognitive science literature on computational models for category learning[66]. However, an assumption typically made by these models is that examples for all categories are either observed together or that they can be directly stored in memory (e.g., exemplar-, prototype- and rule-based models all rely on this assumption). GR could be a biologically plausible way to extend these models to the more natural case in which the different categories to be learned are only available sequentially. Finally, we should also note that there are important aspects of replay in the brain that are missing from our brain-inspired replay method. One of them is temporal structure: replay-events in the brain consist of sequences of neuronal activity that reflect the temporal order of the actual experiences[67]. A recent method for continual learning that incorporates such sequence replay was proposed by Parisi et al.[68]. Their approach has several conceptual similarities with our internal replay component, as they also use pre-trained convolutional layers as a feature extractor and they replay network embeddings rather than pixel-level images. However, an important difference is that their method does not learn an explicit generative model to generate these embeddings, but it stores them using a recurrent self-organizing network. A disadvantage of this approach is that it depends on growing new neurons for each new experience, which raises doubt whether their method could scale to the protocols considered here.

An intriguing question is why GR is so much more effective for Class-IL than regularization-based methods such as EWC and SI. One answer may relate to how these different approaches store and maintain the memory of previously encountered classes. GR maintains this memory in the function or output space of the network, because this method learns to produce input-output combinations that the network should not forget. On the other hand, regularization-based methods store and maintain the memory of previous classes entirely in the parameter space of the network, as their only tool is to vary the plasticity of parameters. Especially with Class-IL this might be challenging, since all information about previous classes must be kept, as it is unknown what the future classes will be like. With Task-IL the memory to be stored is simpler, because only the features important for the specific task learned at that time need to be remembered. Nevertheless, we want to highlight that although we found that current regularization-based methods were not able to solve Class-IL by themselves, they did provide a unique contribution when combined with GR. We hypothesize that this is because maintaining memories in function space and maintaining them in parameter space each come with their own, separate challenges: for GR the challenge is to learn a generative network that captures enough of the essence of the previous tasks/classes, while for regularization-based approaches the challenge is to correctly assign credit to the parameters of the network. This suggests that regularization (or metaplasticity) and replay are complementary mechanisms, which is consistent with empirical observations that the brain uses both strategies—roughly corresponding to cellular

and systems consolidation, respectively—side by side to protect its memories[69].

## Methods

**Task protocols**. For the split MNIST task protocol, we split up the MNIST dataset[33] in five tasks or episodes, such that each task/episode contained two digits. The original $28 \times 28$ pixel grey-scale images were used, no pre-processing was applied. We used the standard training/test-split, resulting in 60,000 training images (~6000 per digit) and 10,000 test images (~1000 per digit).

For permuted MNIST, the original MNIST images were first zero-padded to $32 \times 32$ pixels. For each task, a random permutation was then generated and applied to these 1024 pixels. No other pre-processing was performed. A sequence of 100 tasks was used, whereby each task was a ten-way classification with as goal to identify the original digit. Again the standard training/test-split was used, so that for each task there were 60,000 training images and 10,000 test images.

For split CIFAR-100, the CIFAR-100 dataset[35] was split into ten tasks or episodes, such that each task/episode consisted of ten classes. The original $32 \times 32$ pixel RGB-colour images were normalized (i.e., each pixel-value was subtracted by the relevant channel-wise mean and divided by the channel-wise standard deviation, with means and standard deviations calculated over all training images), but no other pre-processing or augmentation was performed. The standard training/test-split was used, resulting in 50,000 training images (500 per class) and 10,000 test images (100 per class).

**Network architecture**. For a fair comparison, the same base neural network architecture was used for all compared methods. For split MNIST, this was a fully connected network with two-hidden layers of 400 nodes each and a softmax output layer. ReLU non-linearities were used in all hidden nodes. The same network architecture was used for permuted MNIST, except with 2000 nodes per hidden layer, similar as in ref. [31]. For split CIFAR-100, the base neural network architecture consisted of five-pre-trained convolutional layers (see below) followed by two-fully connected layers each containing 2000 nodes with ReLU non-linearities and a softmax output layer.

The softmax output layer was treated differently depending on the scenario according to which a task protocol was performed. In both the Task-IL and the Class-IL scenario, the output layer always had a separate output unit for each class to be learned (i.e., there were 10 output units for split MNIST and 100 for split CIFAR-100), but these scenarios differed in when these output units were 'active'. In the Task-IL scenario the output layer was 'multi-headed', meaning that always only the output units of classes in the task under consideration—i.e., either the current task or the replayed task—were active; while in the Class-IL scenario the output layer was 'single-headed', meaning that always all output units of the classes encountered so far were active. For permuted MNIST, which was performed according to the Domain-IL scenario, the output layer had one unit for each digit (i.e., there were 10 output units) and all output units were active all the time.

Whether an output unit was active in a given task decided whether the network could assign a positive probability to the corresponding class: the normalization performed by the softmax output layer only took into account the active nodes. That is, the by the neural network predicted conditional probability that an input $\mathbf{x}$ belongs to class $c$ was calculated as:

$$p_{\boldsymbol{\theta}}(Y=c|\mathbf{x}) = \begin{cases} \frac{e^{z_c^{(\mathbf{x})}}}{\sum_j e^{z_j^{(\mathbf{x})}}} & \text{if output node } c \text{ is active} \\ 0 & \text{otherwise} \end{cases} \quad (1)$$

whereby $z_c^{(\mathbf{x})}$ is the unnormalized probability or 'logit' of class $c$ obtained by putting input $\mathbf{x}$ through the neural network parameterized by $\boldsymbol{\theta}$ (note that $z_c^{(\mathbf{x})}$ thus also depends on $\boldsymbol{\theta}$, but to lighten notation this dependence is suppressed), and the summation in the denominator is over all active nodes in the output layer.

**Training**. The goal was always to sequentially train the neural network on all tasks or episodes of the task protocol, whereby the network only had access to the data of the current task/episode. For all methods considered in this paper, during training the parameters $\boldsymbol{\theta}$ of the neural network were updated with mini-batch stochastic gradient descent on a task-specific loss function ($\mathcal{L}_{\text{total}}$). The exact formulation of this loss function differed between methods, but a central component was always the standard multi-class cross-entropy classification loss on the data of the current task. For an input $\mathbf{x}$ labelled with a hard target $y$, the per-sample classification loss is given by:

$$\mathcal{L}^{\text{C}}(\mathbf{x}, y; \boldsymbol{\theta}) = -\log p_{\boldsymbol{\theta}}(Y=y|\mathbf{x}), \quad (2)$$

with $p_{\boldsymbol{\theta}}$ the conditional probability distribution defined by the neural network (see Eq. (1)).

Each task was trained for 2000 (split MNIST and permuted MNIST) or 5000 (split CIFAR-100) iterations using the ADAM-optimizer ($\beta_1 = 0.9$, $\beta_2 = 0.999$)[70] with learning rate of 0.001 (split MNIST) or 0.0001 (permuted MNIST and split CIFAR-100). The mini-batch size was set to 128 (split MNIST) or 256 (permuted MNIST and split CIFAR-100), meaning that in each iteration $\mathcal{L}_{\text{current}}$ (see below) was calculated as average over that many samples from the current task. If replay

was used, the same number of replayed samples was used to calculate $\mathcal{L}_{\text{replay}}$ (see below). Only exception to this were the experiments in Fig. 4a, in which the mini-batch size used for replay was systematically varied while the mini-batch size for the current task was kept at 128.

**Baselines**. To assess how successful the compared methods are in alleviating catastrophic forgetting, we included two baselines in each comparison.

For our naive baseline, the base neural network was sequentially trained on all tasks in the standard way (i.e., the loss was always the classification loss on the data of the current task: $\mathcal{L}_{\text{total}} = \mathcal{L}_{\text{current}} = \mathcal{L}^{\text{C}}$).

To get an upper bound, the base neural network was always trained using the training data of all tasks so far ('joint training'). This is also referred to as offline training.

**Main model for GR**. For standard GR, two models were sequentially trained on all tasks: (1) the main model, for actually solving the tasks, and (2) a separate generative model, for generating inputs representative of previously learned tasks.

The main model was a classifier with the base neural network architecture. The loss function used to train this model consisted of two terms: one for the data of the current task and one for the replayed data, with both terms weighted according to how many tasks/episodes the model had seen so far:

$$\mathcal{L}_{\text{total}} = \frac{1}{N_{\text{tasks so far}}} \mathcal{L}_{\text{current}} + (1 - \frac{1}{N_{\text{tasks so far}}}) \mathcal{L}_{\text{replay}}. \quad (3)$$

For standard GR, $\mathcal{L}_{\text{current}}$ and $\mathcal{L}_{\text{replay}}$ were the standard classification loss on the current task data and on the replayed data.

**Generator for GR**. For the generative model, we used a symmetric VAE[28,71]. A VAE consists of an encoder network $q_{\boldsymbol{\phi}}$ mapping an input-vector $\mathbf{x}$ to a vector of stochastic latent variables $\mathbf{z}$, and a decoder network $p_{\boldsymbol{\psi}}$ mapping those latent variables $\mathbf{z}$ to a reconstructed or decoded input-vector $\hat{\mathbf{x}}$. We kept the architecture of both networks similar to the base neural network: for the MNIST-based protocols both the encoder and the decoder were fully connected networks with two-hidden layers containing 400 (split MNIST) or 2000 (permuted MNIST) units with ReLU non-linearity; for split CIFAR-100 the encoder consisted of five pre-trained convolutional layers (see below) followed by two-fully connected layers containing 2000 ReLU units and the decoder had two-fully connected layers with 2000 ReLU units followed by five deconvolutional or transposed convolutional layers (see below). The stochastic latent variable layer $\mathbf{z}$ always had 100 Gaussian units (parameterized by mean $\boldsymbol{\mu}^{(\mathbf{x})}$ and standard deviation $\boldsymbol{\sigma}^{(\mathbf{x})}$, the outputs of the encoder network $q_{\boldsymbol{\phi}}$ given input $\mathbf{x}$) and the prior over them was the standard normal distribution.

Typically, the parameters of a VAE (collected here in $\boldsymbol{\phi}$ and $\boldsymbol{\psi}$) are trained by maximizing a variational lower bound on the evidence (or ELBO), which is equivalent to minimizing the following per-sample loss function for an input $\mathbf{x}$:

$$\begin{aligned} \mathcal{L}^{\text{G}}(\mathbf{x}; \boldsymbol{\phi}, \boldsymbol{\psi}) &= E_{\mathbf{z} \sim q_{\boldsymbol{\phi}}(.|\mathbf{x})}[-\log p_{\boldsymbol{\psi}}(\mathbf{x}|\mathbf{z})] + D_{\text{KL}}(q_{\boldsymbol{\phi}}(.|\mathbf{x})||p(.)) \\ &= \mathcal{L}^{\text{recon}}(\mathbf{x}; \boldsymbol{\phi}, \boldsymbol{\psi}) + \mathcal{L}^{\text{latent}}(\mathbf{x}; \boldsymbol{\phi}), \end{aligned} \quad (4)$$

whereby $q_{\boldsymbol{\phi}}(.|\mathbf{x}) = \mathcal{N}(\boldsymbol{\mu}^{(\mathbf{x})}, \boldsymbol{\sigma}^{(\mathbf{x})2}\mathbf{I})$ is the posterior distribution over the latent variables $\mathbf{z}$ defined by the encoder given input $\mathbf{x}$, $p(.) = \mathcal{N}(0, \mathbf{I})$ is the prior distribution over the latent variables and $D_{\text{KL}}$ is the Kullback-Leibler divergence. With this combination of prior and posterior, it has been shown that the latent variable regularization term $\mathcal{L}^{\text{latent}}$ can be calculated, without having to do estimation, as[28]:

$$\mathcal{L}^{\text{latent}}(\mathbf{x}; \boldsymbol{\phi}) = \frac{1}{2} \sum_{j=1}^{100} \left(1 + \log(\sigma_j^{(\mathbf{x})2}) - \mu_j^{(\mathbf{x})2} - \sigma_j^{(\mathbf{x})2}\right), \quad (5)$$

whereby $\mu_j^{(\mathbf{x})}$ and $\sigma_j^{(\mathbf{x})}$ are the $j^{\text{th}}$ elements of respectively $\boldsymbol{\mu}^{(\mathbf{x})}$ and $\boldsymbol{\sigma}^{(\mathbf{x})}$. To simplify the reconstruction term $\mathcal{L}^{\text{recon}}$, we made the output of the decoder network $p_{\boldsymbol{\psi}}$ deterministic, which is a modification that is common in the VAE literature[72,73]. We redefined the reconstruction term as the expected binary cross entropy between the original and the decoded pixel values:

$$\mathcal{L}^{\text{recon}}(\mathbf{x}; \boldsymbol{\phi}, \boldsymbol{\psi}) = E_{\boldsymbol{\epsilon} \sim \mathcal{N}(\mathbf{0}, \mathbf{I})} \left[ \sum_{p=1}^{N_{\text{pixels}}} x_p \log\left(\hat{x}_p\right) + \left(1 - x_p\right) \log\left(1 - \hat{x}_p\right) \right], \quad (6)$$

whereby $x_p$ was the value of the $p^{\text{th}}$ pixel of the original input image $\mathbf{x}$ and $\hat{x}_p$ was the value of the $p^{\text{th}}$ pixel of the decoded image $\hat{\mathbf{x}} = p_{\boldsymbol{\psi}}(\mathbf{z}^{(\mathbf{x})})$ with $\mathbf{z}^{(\mathbf{x})} = \boldsymbol{\mu}^{(\mathbf{x})} + \boldsymbol{\sigma}^{(\mathbf{x})} \odot \boldsymbol{\epsilon}$ and $\boldsymbol{\epsilon} \sim \mathcal{N}(\mathbf{0}, \mathbf{I})$. Replacing $\mathbf{z}^{(\mathbf{x})}$ by $\boldsymbol{\mu}^{(\mathbf{x})} + \boldsymbol{\sigma}^{(\mathbf{x})} \odot \boldsymbol{\epsilon}$ is known as the 'reparameterization trick'[28] and makes that a Monte Carlo estimate of the expectation in Eq. (6) is differentiable with respect to $\boldsymbol{\phi}$. As is common in the literature, for this estimate we used just a single sample of $\boldsymbol{\epsilon}$ for each datapoint.

To train the generator, the same hyperparameters (i.e., learning rate, optimizer, iterations, batch sizes) were used as for training the main model. Similar to the main model, the generator was also trained with replay (i.e., $\mathcal{L}_{\text{total}}^{\text{G}} = \frac{1}{N_{\text{tasks so far}}} \mathcal{L}_{\text{current}}^{\text{G}} + (1 - \frac{1}{N_{\text{tasks so far}}}) \mathcal{L}_{\text{replay}}^{\text{G}})$.

**Generating a replay sample**. The data to be replayed were produced by first sampling inputs from the generative model, after which those generated inputs were presented to the main model and labelled as the most likely class as predicted by that model (Fig. 2a). There were however two subtleties regarding the generation of these replay samples.

Firstly, the samples replayed during task $T$ were generated by the versions of the generator and main model directly after finishing training on task $T - 1$. Implementationally, this could be achieved either by temporarily storing a copy of both models after finishing training on each task, or by already generating all samples to be replayed during training on an upcoming task before training on that task is started.

Secondly, there were some differences between the three continual learning scenarios regarding which output units were active when generating the labels for the inputs to be replayed (i.e., which classes could be predicted). With the Task-IL scenario, only the output units corresponding to the classes of the task intended to be replayed were active (with the task intended to be replayed randomly selected among the previous tasks), while with the Class-IL scenario the output units of the classes from up to the previously learned task were active. With the Domain-IL scenario, always all output units were active.

**Distillation**. For brain-inspired replay, instead of labelling the inputs to be replayed only with an integer indicating the most likely class as predicted by the main model (i.e., 'hard targets'), we labelled them with a vector containing the predicted probabilities for all possible classes (i.e., 'soft targets'). As is common for distillation, these target probabilities with which the generated inputs were labelled were made 'softer' by raising the temperature $T$ of the softmax layer of the main model when generating the labels. This meant that before the softmax normalization was performed on the logits $\mathbf{z}^{(\mathbf{x})}$ (see Eq. (1)), these logits were first divided by $T$. That is, for an input $\mathbf{x}$ to be replayed during training of task $K$, the soft targets were given by the vector $\tilde{\mathbf{y}}$ with $c^{\text{th}}$ element equal to:

$$\tilde{y}_c = p_{\hat{\boldsymbol{\theta}}^{(K-1)}}^T (Y = c | \mathbf{x}), \tag{7}$$

where $\hat{\boldsymbol{\theta}}^{(K-1)}$ is the vector with parameter values after finishing training on task $K - 1$ and $p_{\boldsymbol{\theta}}^T$ is the conditional probability distribution defined by the neural network with parameters $\boldsymbol{\theta}$ and with the temperature of the softmax layer raised to $T$. A typical value for this temperature is 2, which was the value used here.

The standard classification loss $\mathcal{L}^{\text{C}}$ could not be used for replayed samples labelled with soft targets. Instead, the training objective was to match the probabilities predicted by the model being trained to these soft targets by minimizing the cross entropy between them. For an input $\mathbf{x}$ labelled with a soft target vector $\tilde{\mathbf{y}}$, the per-sample distillation loss is given by:

$$\mathcal{L}^{\text{D}}(\mathbf{x}, \tilde{\mathbf{y}}; \boldsymbol{\theta}) = -T^2 \sum_{c=1}^{N_{\text{classes}}} \tilde{y}_c \log p_{\boldsymbol{\theta}}^T (Y = c | \mathbf{x}), \tag{8}$$

with temperature $T$ again set to 2. The scaling by $T^2$ was included to ensure that the relative contribution of this objective matched that of a comparable objective with hard targets[50].

**Replay-through-feedback**. The RtF model was implemented as a symmetric VAE with an added softmax classification layer to the final hidden layer of the encoder (i.e., the layer before the latent variable layer; Fig. 7a). Now only one model had to be trained. To train this model, the loss function for the data of the current task had two terms that were simply added: $\mathcal{L}_{\text{current}}^{\text{RtF}} = \mathcal{L}^{\text{C}} + \mathcal{L}^{\text{G}}$, whereby $\mathcal{L}^{\text{C}}$ was the standard cross-entropy classification loss (see Eq. (2)) and $\mathcal{L}^{\text{G}}$ was the generative loss (see Eq. (4)). For the replayed data, when distillation was used, the classification term was replaced by the distillation term from Eq. (8): $\mathcal{L}_{\text{replay}}^{\text{RtF}} = \mathcal{L}^{\text{D}} + \mathcal{L}^{\text{G}}$. The loss terms for the current and replayed data were again weighted according to how many tasks/episodes the model had seen so far: $\mathcal{L}_{\text{total}}^{\text{RtF}} = \frac{1}{N_{\text{tasks so far}}} \mathcal{L}_{\text{current}}^{\text{RtF}} + (1 - \frac{1}{N_{\text{tasks so far}}}) \mathcal{L}_{\text{replay}}^{\text{RtF}}$.

**Conditional replay**. To enable the network to generate examples of specific classes, we replaced the standard normal prior over the stochastic latent variables $\mathbf{z}$ by a Gaussian mixture with a separate mode for each class (Fig. 7b):

$$p_{\chi}(.) = \sum_{c=1}^{N_{\text{classes}}} p(Y = c) p_{\chi}(.|c), \tag{9}$$

with $p_{\chi}(.|c) = \mathcal{N}(\boldsymbol{\mu}^c, \boldsymbol{\sigma}^{c2}\boldsymbol{I})$ for $c = 1, \ldots, N_{\text{classes}}$, whereby $\boldsymbol{\mu}^c$ and $\boldsymbol{\sigma}^c$ are the trainable mean and standard deviation of the mode corresponding to class $c$, $\chi$ is the collection of trainable means and standard deviations of all classes and $p(Y = c) = \text{Categorical}\left(\frac{1}{N_{\text{classes}}}\right)$ is the class-prior. Because of the change in prior distribution, the expression for analytically calculating the latent variable regularization term in Eq. (5) is no longer valid. In the Supplementary Methods we show that for an input $\mathbf{x}$ labelled with a hard target $y$ (i.e., for current task data), $\mathcal{L}^{\text{latent}}$ still

has a closed-form expression:

$$\mathcal{L}^{\text{latent}}(\mathbf{x}, y; \boldsymbol{\phi}, \chi) = \frac{1}{2} \sum_{j=1}^{100} \left( 1 + \log\left(\sigma_j^{(\mathbf{x})2}\right) - \log\left(\sigma_j^{y2}\right) - \frac{\left(\mu_j^{(\mathbf{x})} - \mu_j^y\right)^2 + \sigma_j^{(\mathbf{x})2}}{\sigma_j^{y2}} \right), \tag{10}$$

whereby $\mu_j^y$ and $\sigma_j^y$ are the $j^{\text{th}}$ elements of respectively $\boldsymbol{\mu}^y$ and $\boldsymbol{\sigma}^y$. However, for an input $\mathbf{x}$ labelled with a soft target $\tilde{\mathbf{y}}$ (i.e., for replayed data), there is no closed-form expression for $\mathcal{L}^{\text{latent}}$ so we resort to estimation by sampling. In the Supplementary Methods we show that in the case of soft targets, $\mathcal{L}^{\text{latent}}$ can be rewritten as:

$$
\begin{aligned}
\mathcal{L}^{\text{latent}}(\mathbf{x}, \tilde{\mathbf{y}}; \boldsymbol{\phi}, \chi) = &\frac{1}{2} \sum_{j=1}^{100} \left( 1 + \log\left(2\pi\right) + \log\left(\sigma_j^{(\mathbf{x})2}\right) \right) \\
&+ E_{\boldsymbol{\epsilon} \sim \mathcal{N}(\mathbf{0}, \boldsymbol{I})} \left[ \log\left( \sum_{j=1}^{100} \tilde{y}_j \mathcal{N}\left(\boldsymbol{\mu}^{(\mathbf{x})} + \boldsymbol{\sigma}^{(\mathbf{x})} \odot \boldsymbol{\epsilon} | \boldsymbol{\mu}^j, \boldsymbol{\sigma}^{j2}\boldsymbol{I}\right) \right) \right],
\end{aligned}
\tag{11}
$$

whereby $\tilde{y}_j$ is the $j^{\text{th}}$ element of $\tilde{\mathbf{y}}$. The expectation in Eq. (11) was estimated with a single Monte Carlo sample per datapoint.

When generating a sample to be replayed, the specific class $y$ to be generated was first randomly selected from the classes seen so far, after which the latent variables $\mathbf{z}$ were sampled from $\mathcal{N}(\boldsymbol{\mu}^y, \boldsymbol{\sigma}^{y2}\boldsymbol{I})$. Although this meant that a specific class was intended to be replayed (and that class could thus be used to label the generated sample with a hard target), it was still the case that the generated inputs were labelled based on the predictions made for them by a feedforward pass through the (main) model.

**Gating based on internal context**. To enable context-dependent processing in the generative part of our models, for each task or class to be learned, a randomly selected subset of $X$% of the units in each hidden layer of the decoder network was fully gated (i.e., their activations were set to zero; Fig. 7c). There was a different mask either for each task (permuted MNIST) or for each class (CIFAR-100) to be learned. As for the original version of XdG, $X$ was a hyperparameter whose value was set by a grid search (Supplementary Figs. S2 and S3). When combined with conditional replay, during the generation of the samples to be replayed, the specific classes selected to be generated dictated which task- or class-masks to use. When not combined with conditional replay, the task- or class-masks to use when generating replay were selected randomly from the tasks or classes seen so far.

**Internal replay**. To achieve the replay of hidden or internal representations, we removed the deconvolutational or transposed convolutional layers from the decoder network. During reconstruction or generation, samples thus only passed through the fully connected layers of the decoder. This meant that replayed samples were generated at an intermediate or 'hidden' level, and those replayed intermediate representations then entered the encoder network after the convolutional layers (Fig. 7d). The reconstruction term of the generative part of the loss function was therefore changed from the input level to the hidden level, and it was defined as the expected squared error between the hidden activations of the original input and the corresponding hidden activations after decoding:

$$\mathcal{L}^{\text{i-recon}}(\mathbf{x}; \boldsymbol{\phi}, \boldsymbol{\psi}) = E_{\boldsymbol{\epsilon} \sim \mathcal{N}(\mathbf{0}, \boldsymbol{I})} \left[ \sum_{i=1}^{N_{\text{units}}} \left( h_i^{(\mathbf{x})} - \hat{h}_i \right)^2 \right], \tag{12}$$

whereby $h_i^{(\mathbf{x})}$ was the activation of the $i^{\text{th}}$ hidden unit when the original input image $\mathbf{x}$ was put through the convolutional layers, and $\hat{h}_i$ was the activation of the $i^{\text{th}}$ hidden unit after decoding the original input: $\hat{\boldsymbol{h}} = p_{\boldsymbol{\psi}}^*(\mathbf{z}^{(\mathbf{x})})$ with $p_{\boldsymbol{\psi}}^*$ the decoder network without deconvolutational layers and $\mathbf{z}^{(\mathbf{x})} = \boldsymbol{\mu}^{(\mathbf{x})} + \boldsymbol{\sigma}^{(\mathbf{x})} \odot \boldsymbol{\epsilon}$ with $\boldsymbol{\epsilon} \sim \mathcal{N}(\mathbf{0}, \boldsymbol{I})$. The expectation in Eq. (12) was again estimated with a single Monte Carlo sample per datapoint. To prevent large changes to the convolutional layers of the encoder, the convolutional layers were pre-trained on CIFAR-10 (see below) and frozen during the incremental training on CIFAR-100. For a fair comparison, for split CIFAR-100, pre-trained convolutional layers were used for all compared methods. Note that with the permuted MNIST protocol the base network did not contain any convolutional layers; for this protocol internal replay and pre-training were therefore not used.

**Addition- and ablation-experiments**. The five components of brain-inspired replay—RtF, conditional replay, gating based on internal context, internal replay and distillation—are modular and they could be used independent of each other. This property was used for the analyses in Figs. 8 and 9 aimed at tearing apart the separate and combined contributions of these modifications.

**Pre-trained convolutional layers**. For split CIFAR-100, all networks had five convolutional layers containing 16, 32, 64, 128 and 254 channels. Each layer used a $3 \times 3$ kernel, a padding of 1, and there was a stride of 1 in the first layer (i.e., no downsampling) and a stride of 2 in the other layers (i.e., image-size was halved in

each of those layers). All convolutional layers used batch-norm[74] followed by a ReLU non-linearity. For the $32 \times 32$ RGB pixel images used in this study, these convolutional layers returned $256 \times 2 \times 2 = 1024$ image features. No pooling was used. Mirroring these convolutional layers, the decoder network of the VAE that was used with standard GR had five deconvolutional or transposed convolutional layers[75] containing 128, 64, 32, 16 and 3 channels. The first four deconvolutional layers used a $4 \times 4$ kernel, a padding of 1 and a stride of 2 (i.e., image-size was doubled in each of those layers), while the final layer used a $3 \times 3$ kernel, a padding of 1 and a stride of 1 (i.e., no upsampling). The first four deconvolutional layers used batch-norm followed by a ReLU non-linearity, while the final layer had no non-linearity.

To simulate development, the convolutional layers were pre-trained on CIFAR-10, which is a dataset containing similar but non-overlapping images and image classes compared to CIFAR-100[35]. Pre-training was done by training the base neural network to classify the 10 classes of CIFAR-10 for 100 epochs, using the ADAM-optimizer ($\beta_1 = 0.9$, $\beta_2 = 0.999$) with learning rate of 0.0001 and mini-batch size of 256.

**Regularization-based methods**. For the regularization-based methods (SI[29], EWC[25] and online EWC[7]), to penalize changes to parameters important for previously learned tasks, a regularization term was added to the loss: $\mathcal{L}_{\text{total}} = \mathcal{L}_{\text{current}} + \lambda \mathcal{L}_{\text{regularization}}$. The value of hyperparameter $\lambda$ was set by a grid search (see Supplementary Figs. 1–3; for SI this hyperparameter is typically referred to as $c$).

For SI, to estimate the importance of parameter $i$, after every task $k$ the contribution of that parameter to the change in loss was first calculated as:

$$\omega_i^{(k)} = \sum_{t=1}^{N_{\text{iters}}} \left( \hat{\theta}_i[t^{(k)}] - \hat{\theta}_i[(t-1)^{(k)}] \right) \frac{-\delta \mathcal{L}_{\text{total}}[t^{(k)}]}{\delta \theta_i}, \quad (13)$$

with $N_{\text{iters}}$ the number of iterations per task, $\hat{\theta}_i[t^{(k)}]$ the value of the $i^{\text{th}}$ parameter after the $t^{\text{th}}$ training iteration on task $k$ and $\frac{\delta \mathcal{L}_{\text{total}}[t^{(k)}]}{\delta \theta_i}$ the gradient of the loss with respect to the $i^{\text{th}}$ parameter during the $t^{\text{th}}$ training iteration on task $k$. To get the estimated importance of parameter $i$ for the first $K-1$ tasks, these contributions were normalized by the square of the total change of parameter $i$ during training on that task plus a small dampening term $\xi$ (set to 0.1, to bound the normalized contributions when a parameter's total change goes to zero), after which they were summed over all tasks so far:

$$\Omega_i^{(K-1)} = \sum_{k=1}^{K-1} \frac{\omega_i^{(k)}}{\left( \Delta_i^{(k)} \right)^2 + \xi}, \quad (14)$$

with $\Delta_i^{(k)} = \hat{\theta}_i[N_{\text{iters}}^{(k)}] - \hat{\theta}_i[0^{(k)}]$, where $\hat{\theta}_i[0^{(k)}]$ indicates the value of parameter $i$ right before starting training on task $k$. The regularization term of SI during training on task $K > 1$ was then given by:

$$\mathcal{L}_{\text{regularization}_{\text{SI}}}^{(K)}(\boldsymbol{\theta}) = \sum_{i=1}^{N_{\text{params}}} \Omega_i^{(K-1)} \left( \theta_i - \hat{\theta}_i^{(K-1)} \right)^2, \quad (15)$$

whereby $\hat{\theta}_i^{(K-1)}$ is the value of parameter $i$ after finishing training on task $K-1$.

For EWC and online EWC, the importance of parameter $i$ for task $k$ was estimated by the $i^{\text{th}}$ diagonal element of that task's Fisher Information matrix, evaluated at the optimal parameter values after finishing training on that task:

$$F_{ii}^{(k)} = \frac{1}{|S^{(k)}|} \sum_{\mathbf{x} \in S^{(k)}} \left( \sum_{c=1}^{N_{\text{classes}}} \tilde{y}_c^{(\mathbf{x})} \left( \frac{\delta \log p_{\boldsymbol{\theta}}(Y = c | \mathbf{x})}{\delta \theta_i} \Big|_{\boldsymbol{\theta} = \hat{\boldsymbol{\theta}}^{(k)}} \right)^2 \right), \quad (16)$$

with $S^{(k)}$ the training data of task $k$, $\hat{\boldsymbol{\theta}}^{(k)}$ the vector with parameter values after finishing training on task $k$, $p_{\boldsymbol{\theta}}$ the conditional probability distribution defined by the neural network with parameters $\boldsymbol{\theta}$ and $\tilde{y}_c^{(\mathbf{x})} = p_{\hat{\boldsymbol{\theta}}^{(k)}}(Y = c | \mathbf{x})$. For EWC, the regularization term during training on task $K > 1$ was then given by:

$$\mathcal{L}_{\text{regularization}_{\text{EWC}}}^{(K)}(\boldsymbol{\theta}) = \sum_{k=1}^{K-1} \left( \frac{1}{2} \sum_{i=1}^{N_{\text{params}}} F_{ii}^{(k)} \left( \theta_i - \hat{\theta}_i^{(k)} \right)^2 \right). \quad (17)$$

For online EWC, the regularization term during training on task $K > 1$ was given by:

$$\mathcal{L}_{\text{regularization}_{\text{online-EWC}}}^{(K)}(\boldsymbol{\theta}) = \sum_{i=1}^{N_{\text{params}}} \tilde{F}_{ii}^{(K-1)} \left( \theta_i - \hat{\theta}_i^{(K-1)} \right)^2, \quad (18)$$

whereby $\tilde{F}_{ii}^{(K-1)}$ is a running sum of the $i^{\text{th}}$ diagonal elements of the Fisher Information matrices of the first $K-1$ tasks, with a hyperparameter $\gamma \leq 1$ that governs a gradual decay of the contributions of previous tasks. That is: $\tilde{F}_{ii}^{(k)} = \gamma \tilde{F}_{ii}^{(k-1)} + F_{ii}^{(k)}$, with $\tilde{F}_{ii}^{(1)} = F_{ii}^{(1)}$. We always set $\gamma$ to 1. Note that the version of EWC used by Masse et al.[31], for which the results are discussed in the main text, was in fact online EWC with $\gamma$ equal to 1. For split MNIST and split CIFAR-100,

the performances of EWC and online EWC were almost identical, so for those task protocols the results of online EWC were omitted.

**Context-dependent gating**. For XdG[31], the base neural network was sequentially trained on all tasks using the standard loss (i.e., $\mathcal{L}_{\text{total}} = \mathcal{L}_{\text{current}} = \mathcal{L}^C$). Only difference was that for each task a different, randomly selected subset of $X\%$ of the units in every fully connected hidden layer was inhibited (i.e., their activations were set to zero), with $X$ a hyperparameter whose value was set by a grid search (Supplementary Figs. 1 and 3). A limitation of this method is that it requires availability of task identity at test time; it could therefore only be used in the Task-IL scenario.

**Learning without forgetting**. LwF[32] was implemented similarly to standard GR, but instead of replaying generated inputs, the inputs of the current task were replayed. So there was no need to train a generator. This method further used distillation to label the replayed samples. That is, the inputs from the current task were replayed after being labelled with soft targets provided by a copy of the model stored after finishing training on the previous task.

**Measures for evaluating generator performance**. To quantify the quality and diversity of the generated samples replayed by different variants of GR in the Class-IL scenario of split CIFAR-100, we reported modified versions of IS[51], FID[52] and the Precision & Recall curves[53]. The first step of the original versions of these measures is to embed samples into a different space using Inception Net[54]. However, as samples generated at the internal level could not be fed into this network, we replaced the Inception Net by a different neural network (see also ref. [55]). The neural network we used had the same architecture as the base neural network: five pre-trained convolutional layers (as above) followed by two-fully connected layers each containing 2000 nodes with ReLU non-linearities and a softmax output layer with 100 nodes. With the convolutional layers frozen, this network was trained offline to classify the 100 classes of CIFAR-100 for 20 epochs, using the ADAM-optimizer ($\beta_1 = 0.9$, $\beta_2 = 0.999$) with learning rate of 0.0001 and mini-batch size of 256. Because this network had the same pre-trained convolutional layers as the encoder of the VAE, samples generated at the internal level could be fed into this network after those convolutional layers. Except for the use of this different embedding network, the three measures were computed as they were originally proposed.

The modified IS measure was calculated by first putting $N_{\text{gen}} = 10,000$ generated samples through the embedding network to obtain for each generated sample $\mathbf{x}_g$ a conditional probability distribution $p(. | \mathbf{x}_g)$ over the 100 classes of the CIFAR-100 dataset. The modified IS was then calculated as:

$$\text{mIS} = \exp \left( \frac{1}{N_{\text{gen}}} \sum_{g=1}^{N_{\text{gen}}} D_{\text{KL}} \left( p(. | \mathbf{x}_g) || p(.) \right) \right), \quad (19)$$

whereby $p(.) = \frac{1}{N_{\text{gen}}} \sum_{g=1}^{N_{\text{gen}}} p(. | \mathbf{x}_g)$. Because a 100-way classifier is used as embedding network, this modified IS measure is bounded between 1 and 100 (see Appendix of ref. [76]).

To calculate the modified FID measure, the 10,000 real data samples from the held-out test set and 10,000 generated samples were first transformed to feature vectors by putting them through all but the final softmax layer of the embedding network. The modified FID was then calculated as:

$$\text{mFID} = ||\boldsymbol{\mu}_r - \boldsymbol{\mu}_g||_2^2 + \text{Tr} \left( \Sigma_r + \Sigma_g - 2 \left( \Sigma_r \Sigma_g \right)^{\frac{1}{2}} \right), \quad (20)$$

whereby $(\boldsymbol{\mu}_r, \Sigma_r)$ and $(\boldsymbol{\mu}_g, \Sigma_g)$ are the mean and covariance of the feature vectors of the real and generated data. This measure is bounded below by 0 and has no maximum value.

For the Precision & Recall curve, the first step was again to transform the 10,000 samples from the test set and 10,000 generated samples to feature vectors by putting them through all but the final softmax layer of the embedding network. The resulting 20,000 feature vectors were then divided into 20 clusters using $k$-means clustering. Letting $P(\omega)$ and $Q(\omega)$ be the proportion of real and generated feature vectors in cluster $\omega$, the Precision & Recall curve was then calculated as:

$$\text{PR} = \{ (\alpha(\lambda), \beta(\lambda)) | \lambda \in \Lambda \}, \quad (21)$$

whereby $\alpha(\lambda) = \sum_{\omega=1}^{20} \min(\lambda P(\omega), Q(\omega))$ is the precision, $\beta(\lambda) = \sum_{\omega=1}^{20} \min(P(\omega), \frac{Q(\omega)}{\lambda})$ is the recall and $\Lambda = \left\{ \tan \left( \frac{i}{m+1} \frac{\pi}{2} \right) | i = 1, 2, ..., m \right\}$. Both the precision $\alpha$ and the recall $\beta$ are bounded between 0 and 1. The clustering step and above computation were repeated 10 times and the resulting curves were averaged.

In addition to these three measures, we also reported the average estimated log-likelihood and the reconstruction error (see Supplementary Fig. 4), although it should be noted that comparing between samples generated at the internal level versus at the pixel level might not be fair with these measures (see Supplementary Methods).

## Data availability

All datasets analysed in this study are freely available online resources.

## Code availability

Detailed, well-documented code that can be used to reproduce or build upon the reported experiments is freely available online under an MIT License: https://github.com/GMvandeVen/brain-inspired-replay.

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

## Acknowledgements

We thank Mengye Ren, Zhe Li and Máté Lengyel for comments on various parts of this work, and Johannes Oswald and Zhengwen Zeng for useful suggestions. This research project has been supported by an IBRO-ISN Research Fellowship, by the Lifelong Learning Machines (L2M) program of the Defence Advanced Research Projects Agency (DARPA) via contract number HR0011-18-2-0025 and by the Intelligence Advanced Research Projects Activity (IARPA) via Department of Interior/Interior Business Center (DoI/IBC) contract number D16PC00003. The views and conclusions contained herein are those of the authors and should not be interpreted as necessarily representing the official policies or endorsements, either expressed or implied, of DARPA, IARPA, DoI/IBC, or the U.S. Government.

## Author contributions

Conceptualization, G.M.v.d.V, H.T.S. and A.S.T.; Formal analysis, G.M.v.d.V.; Funding acquisition, A.S.T. and G.M.v.d.V.; Investigation, G.M.v.d.V.; Methodology, G.M.v.d.V.; Resources, A.S.T.; Software, G.M.v.d.V.; Supervision, A.S.T.; Visualization, G.M.v.d.V.; Writing – original draft, G.M.v.d.V.; Writing – review and editing, G.M.v.d.V., H.T.S. and A.S.T.

## Competing interests

The authors declare no competing interests.
