## [Peer Review File · Nature Communications]

Reviewers' comments:

Reviewer #1 (Remarks to the Author):

After carefully evaluating a number of approaches to prevent catastrophic forgetting in neural networks during sequential learning trials, the authors propose a "brain-like" approach and demonstrate that it obtains state-of-the-art performance on challenging natural image sequences. The principal contributions of the paper are twofold: to provide an insightful discussion and examination of existing continual learning methods, and to motivate, describe, and evaluate a novel approach.

The paper is well-written, insightful, and timely, given the increased interest in catastrophic forgetting by the AI/ML community in recent years. The authors have chosen a good representative set of baseline algorithms to ground their empirical study: EWC, SI, Generative replay, XdG, and LWF. After identifying class-incremental learning as the more challenging scenario, the different methods are compared on MNIST tasks. The authors then describe the innovative aspects of their proposed 'brain-like' approach and show that it does substantially better on CIFAR100 in the class-incremental setting.

Although the empirical comparison of baseline algorithms is interesting to read and provides a good introduction to the field, it is narrowly focused on MNIST classification as a sole problem domain (both split MNIST and permuted MNIST are evaluated). There is not even a mention of RL or unsupervised sequential learning environments. Moreover, the paper purports to be about continual learning, but reduces the topic to just catastrophic forgetting, while ignoring problems like interference and forward transfer that are critical aspects of continual learning. When the paper is revised, the authors should clearly establish the limitations and assumptions that they are making in this regard.

Most important, however, is to understand the value of the algorithmic contribution offered by the paper. The 'brain-inspired replay' approach proposes five modifications to the generative replay approach in (Shin et al). First, they use soft rather than hard targets when replaying generated samples, similar to distillation. This is actually presented as a baseline, but I'm not sure why the authors did this: it would have been better to see the actual (Shin et al) method evaluated and then see the distillation comparison, either separately or as part of the 'brain-inspired replay'. The other 4 modifications are 1) replay-through-feedback, 2) turning the latent variables into a Gaussian mixture so that samples from different classes can be conditionally generated, 3) gating (as was done in XdG) the generator based on the class of the generated sample, and 4) using pre-trained and frozen convolutional layers and only replaying generated hidden states rather than generated inputs.

None of these modifications is novel, as all are derived from other published approaches, which would usually detract substantially from the merit of the submission. However, in this case the 4 (or 5, if you count the soft targets) modifications do seem to result in a substantially better outcome: a model that is able to learn 100 classes of CIFAR sequentially, without catastrophic forgetting. Moreover, the modifications hang together well; they are complementary and seamless rather than complex or redundant. Unfortunately, it is impossible to support the paper for submission given that the modifications are not separately evaluated. Because the modifications are complementary, it is clearly feasible to ablate the approach and establish the relative importance of each modification, but the authors have not done so. This makes it very unclear whether all of the modifications were needed or even significant for the final performance. Since the modifications are themselves not novel, it is important to show that it is their sum that yields such strong results.

I can't recommend acceptance of the paper as is. I recommend that the authors provide ablations or other empirical evidence to support their method and resubmit when this is completed.

Reviewer #2 (Remarks to the Author):

The authors propose a continual learning approach with memory replay that mitigates catastrophic forgetting in incremental learning tasks. The replay strategy functionally resembles hippocampal replay in the brain in the sense that rather than replaying stored input data or generative versions of it (two techniques also known in the machine learning literature as rehearsal and generative replay respectively), use network embeddings with context-modulated connections.

The topics of lifelong/continual learning are extremely interesting and have received increasing attention in the machine learning literature as well as in neuroscience-inspired computational models of memory and learning. Although I do not find any objective errors or significant flaws in the proposed methodology, I have a series of both major and minor concerns.

1) A major concern regards the extent to which this proposed approach actually resembles enough (at least functionally) biological experience replay to be called "brain-like". The authors' review of replay in the brain is particularly limited and overlooks important factors such as intervals of hippocampal reactivation for replay and the crucial interplay of neocortex and hippocampus for the selection of experience replay (Gelbard-Sagiv et al. 2008; see Carr et al., 2011 for a review). While I agree with the authors that the proposed replay strategy is more biologically plausible than the explicit storage of raw input data or generated input samples for their subsequent replay, I cannot agree on calling this model brain-like because it uses hidden representations.

2) Also, the fact that the proposed method uses pre-trained convolutional layers to "simulate development" is an important oversimplification that raises a number of questions regarding the ability of the model itself to learn continually. In particular, MNIST (the dataset to be learned) is simpler than CIFAR-10 (used as pre-training). This suggests that without this pre-training stage, the model might not be able to learn. The authors also use CIFAR-100 for training that, although more complex than CIFAR-10, does not show whether the model can in fact learn out-of-distribution input. The authors should discuss or empirically show these aspects of the model.

3) The overall approach reminds me of Parisi et al. (2018) in which the authors also 1) use pre-trained convolutional networks as a low-level layer, 2) use network embeddings rather than raw input or pseudo-input for experience replay, 3) do not require task labels. Because of such conceptual analogies, it would be convenient if the authors could explain the potential similarities and differences with this approach. Furthermore, an interesting aspect of this model is the use of temporal context to improve learning. Although the learning of temporally correlated data is probably out of the scope of this study, the authors should speculate whether and how the proposed brain-like model can extend to the temporal domain, a crucial property of biological learning.

4) Of minor concern regards the lack of clarification of some used terminology. For instance, the authors occasionally refer to "true lifelong learning". Although in the ML/DL literature "continual" and "lifelong" are used interchangeably, these two terms have different meanings in other fields such as robotics and behavioral psychology. The authors should clarify what they mean by true lifelong learning.

Carr et al. (2011). Hippocampal replay in the awake state: a potential substrate for memory consolidation and retrieval. *Nat. Neurosci.* 14, 147–153.

Gelbard-Sagiv et al. (2008). Internally generated reactivation of single neurons in human

hippocampus during free recall. *Science* 322, 96–101.

Parisi et al. (2018) Lifelong Learning of Spatiotemporal Representations With Dual-Memory Recurrent Self-Organization. *Front. Neurobot.* 12:78.

Reviewer #3 (Remarks to the Author):

In the manuscript entitled "Brain-like replay for continual learning with artificial neural networks," the authors tackle the problem of catastrophic forgetting in the task-incremental continual learning setting in which a classifier network has a single readout head, which has to learn an increasing number of classes. To that end, the authors propose a continual learning framework that combines context-dependent gating (XdG), VAE-based replay mechanisms, internal replay, and additional parameter-based regularization techniques such as EWC and SI. The authors discuss these modifications in the context of biological plausibility and demonstrate competitive performance on a range of classification tasks.

The manuscript tackles the exciting problem of continual learning in deep neural networks. It is well written with neat figures and correctly cites the relevant literature. Overall, the results seem appealing for a broader community with a primary interest in deep learning models. The claims about "brain-like" mechanisms, however, seem a little bold, too bold perhaps, and could use better support or else could be moderated.

Main points

Statements about brain-like replay

The manuscript makes bold claims about the involved mechanisms being more brain-like, by incorporating a range of mechanisms pertaining to memory replay and context in a supervised deep learning paradigm. However, at the core, the models studied are deep learning models, and there is no explicit connection or justification for why the present proposal is more brain-like. The current presentation seems misleading in this regard, which risks obscuring, perhaps too much, the underlying premise of this work. To alleviate this shortcoming, one would wish for a more rigorous discussion of experimental predictions of the proposed replay mechanism or offer a comparison to experimental data. Overall, it would seem wise to moderate the statements about the "brain-like replay" mechanism. This cut is not to say that the present work is not exciting or brain-inspired.

Contribution of different components to the overall performance

The present model makes several extensions and modifications to published replay mechanisms and illustrates their overall positive net effect on continual learning in the incremental class setting. To avoid the impression of a "bag of hacks", what is presently missing is a fine-grained analysis of which model component contributes what to the overall improvements. For instance, it would be absolutely desirable to see a quantification of the VAE generator performance using one or several established metrics from the VAE literature to measure both sample quality and diversity instead of merely stating the performance of the overall algorithm. The samples shown in Figure 6d do indeed look "too low", as stated in the manuscript, but it should be quantified. Ideally, such quantification would distinguish between the contributions of the multi-modal Gaussian mixture prior, the context-dependent gating in the feedback connections, and both together to quantify how the presumably improved sample quality improves continual learning performance.

Similarly, the difference between internal replay and vanilla (external?) replay should be quantified. Is this a trick that improves performance? Is this done because it seems more biologically plausible, and

it doesn't hurt performance? These questions do not seem addressed in the present version.

Minor suggestions

It isn't entirely clear how XdG context modulation in the generator pass, but not the forward pass, fits with the sought after biological plausibility.

Presently the manuscript dedicates a large amount of "real estate" on the justification that replay might be crucially important. While this is an important point, it puts the main findings of the proposed model relatively late in the manuscript Fig. 5-7. To captivate a broader audience, it could be a good idea to shorten the motivation section and put a stronger emphasis on the actual solution and its components (also by dissecting it; see major point above).

Figure 7: One could consider adding arrows for the generator labels from the Gaussian mixture model.

Point-by-point response to referees' comments

We thank all three referees for their thorough reviews that allowed for a clear improvement of our study. In the revised version of the manuscript, we addressed all the referees' concerns by performing the requested analyses, by discussing the scope and limitations of our study, by relating our work to existing literature and by moderating our claim about the relation between replay in the brain and our proposed modifications to the generative replay method. Below we show the referees' comments (in *italics*), interleaved with our responses.

Referee #1:

After carefully evaluating a number of approaches to prevent catastrophic forgetting in neural networks during sequential learning trials, the authors propose a "brain-like" approach and demonstrate that it obtains state-of-the-art performance on challenging natural image sequences. The principal contributions of the paper are twofold: to provide an insightful discussion and examination of existing continual learning methods, and to motivate, describe, and evaluate a novel approach.

The paper is well-written, insightful, and timely, given the increased interest in catastrophic forgetting by the AI/ML community in recent years. The authors have chosen a good representative set of baseline algorithms to ground their empirical study: EWC, SI, Generative replay, XdG, and LWF. After identifying class-incremental learning as the more challenging scenario, the different methods are compared on MNIST tasks. The authors then describe the innovative aspects of their proposed 'brain-like' approach and show that it does substantially better on CIFAR100 in the class-incremental setting.

Although the empirical comparison of baseline algorithms is interesting to read and provides a good introduction to the field, it is narrowly focused on MNIST classification as a sole problem domain (both split MNIST and permuted MNIST are evaluated). There is not even a mention of RL or unsupervised sequential learning environments. Moreover, the paper purports to be about continual learning, but reduces the topic to just catastrophic forgetting, while ignoring problems like interference and forward transfer that are critical aspects of continual learning. When the paper is revised, the authors should clearly establish the limitations and assumptions that they are making in this regard.

We thank the referee for their positive comments. To address the referee's first concern, in the revised manuscript we now clearly state the scope of our study at the beginning of the results-section: that we focus on image classification problems and that we evaluate performance by looking at overall accuracy, which is a measure that mainly reflects catastrophic forgetting. In the discussion we further added a new section "Limitations and scope", in which we discuss these and other limitations in more details.

Most important, however, is to understand the value of the algorithmic contribution offered by the paper. The 'brain-inspired replay' approach proposes five modifications to the generative replay approach in (Shin et al). First, they use soft rather than hard targets when replaying generated samples, similar to distillation. This is actually presented as a baseline, but I'm not sure why the authors did this: it would have been better to see the actual (Shin et al) method evaluated and then see the distillation comparison, either separately or as part of the 'brain-inspired replay'.

We followed the suggestion of the referee and in our revised manuscript we have changed the definition of "standard generative replay" to no longer include distillation. This means

that we re-did the analyses for the red curves in Figs. 3-6 and the brown curves in Fig. 4. As expected, leaving out distillation reduced performance, but not so much that it affected our interpretation of the results or conclusions. Note that the effect of including or excluding distillation is quantified as part of the addition- and ablation-experiments described below.

We now introduce distillation as a fifth “machine learning inspired” component of our brain-inspired replay method. We rearranged and rewrote the relevant parts of the methods-section to reflect this change.

The other 4 modifications are 1) replay-through-feedback, 2) turning the latent variables into a Gaussian mixture so that samples from different classes can be conditionally generated, 3) gating (as was done in XdG) the generator based on the class of the generated sample, and 4) using pre-trained and frozen convolutional layers and only replaying generated hiddens rather than generated inputs.

None of these modifications is novel, as all are derived from other published approaches, which would usually detract substantially from the merit of the submission. However, in this case the 4 (or 5, if you count the soft targets) modifications do seem to result in a substantially better outcome: a model that is able to learn 100 classes of CIFAR sequentially, without catastrophic forgetting. Moreover, the modifications hang together well; they are complementary and seamless rather than complex or redundant. Unfortunately, it is impossible to support the paper for submission given that the modifications are not separately evaluated. Because the modifications are complementary, it is clearly feasible to ablate the approach and establish the relative importance of each modification, but the authors have not done so. This makes it very unclear whether all of the modifications were needed or even significant for the final performance. Since the modifications are themselves not novel, it is important to show that it is their sum that yields such strong results.

I can't recommend acceptance of the paper as is. I recommend that the authors provide ablations or other empirical evidence to support their method and resubmit when this is completed.

In the revised manuscript we have included the requested ablation-experiments (right side of Fig. 8a,b,c), as well as a complementary series of addition-experiments (left side of Fig. 8a,b,c). These new experiments confirmed that there is an added value in the combination of our modifications: for both permuted MNIST and class-incremental learning on CIFAR-100, the gain in performance obtained by combining all modifications together was larger than the sum of the effects of adding each of them in isolation. In line with this, we also found that for both of these problems, none of the individual modifications were sufficient for the final performance, while all of them except replay-through-feedback were necessary. The contribution of replay-through-feedback is rather to increase efficiency (i.e., no need for two separate models) without substantially hurting performance, although ablating replay-through-feedback sometimes slightly reduced performance as well.

Interestingly, in contrast to class-incremental learning, we found that for task-incremental learning on CIFAR-100 none of the individual modifications were necessary for the final performance. This result fits well with our claim that task-incremental learning is a substantially simpler problem than class-incremental learning.

Finally, to provide further insight into the separate and combined contributions of our modifications, for class-incremental learning on CIFAR-100 we also quantified the quality and diversity of the generated samples replayed by our model both with and without each of our modifications (Fig. 9; see also our response to the third point of referee #3).

Referee #2:

The authors propose a continual learning approach with memory replay that mitigates catastrophic forgetting in incremental learning tasks. The replay strategy functionally resembles hippocampal replay in the brain in the sense that rather than replaying stored input data or generative versions of it (two techniques also known in the machine learning literature as rehearsal and generative replay respectively), use network embeddings with context-modulated connections.

The topics of lifelong/continual learning are extremely interesting and have received increasing attention in the machine learning literature as well as in neuroscience-inspired computational models of memory and learning. Although I do not find any objective errors or significant flaws in the proposed methodology, I have a series of both major and minor concerns.

1) A major concern regards the extent to which this proposed approach actually resembles enough (at least functionally) biological experience replay to be called "brain-like". The authors' review of replay in the brain is particularly limited and overlooks important factors such as intervals of hippocampal reactivation for replay and the crucial interplay of neocortex and hippocampus for the selection of experience replay (Gelbard-Sagiv et al. 2008; see Carr et al., 2011 for a review). While I agree with the authors that the proposed replay strategy is more biologically plausible than the explicit storage of raw input data or generated input samples for their subsequent replay, I cannot agree on calling this model brain-like because it uses hidden representations.

We thank the referee for their insightful comments. We accept the referee's concern about whether our proposed approach resembles enough of biological replay to be called "brain-like". In the revised manuscript we no longer refer to our method as brain-like.

We now only make a claim about our method being brain-inspired. To better motivate why our proposed modifications were inspired by the brain, at various places throughout the manuscript (in the introduction as well as in the results-section) we have extended our review of replay in the brain.

Finally, we also expanded the discussion about the extent to which our proposed method resembles replay in the brain. In particular, we now highlight that there are important aspects of replay in the brain that are absent from our method, such as the temporal organization of replay in the brain.

2) Also, the fact that the proposed method uses pre-trained convolutional layers to "simulate development" is an important oversimplification that raises a number of questions regarding the ability of the model itself to learn continually. In particular, MNIST (the dataset to be learned) is simpler than CIFAR-10 (used as pre-training). This suggests that without this pre-training stage, the model might not be able to learn. The authors also use CIFAR-100 for training that, although more complex than CIFAR-10, does not show whether the model can in fact learn out-of-distribution input. The authors should discuss or empirically show these aspects of the model.

We should start by pointing out that pre-training was actually not used for our experiments with MNIST (i.e., split MNIST and permuted MNIST). We appreciate that this was not clearly stated (it was only mentioned in the methods-section in an admittedly round-about way), and in the revised manuscript we have made this more explicit in both the main text and in the methods-section. For the experiments on MNIST it is thus clear that our model itself was able to learn continually.

For the split CIFAR-100 experiments we did use pre-trained convolutional layers, and we agree with the referee that this is a highly simplified way to simulate development. In the revised manuscript we now discuss the limitations of this aspect of our model in the new section "Limitations and scope" in the discussion. In particular, we discuss that because of the pre-trained convolutional layers it is unlikely that our model could learn out-of-

distribution inputs such as images without natural image statistics. We also point out that because of the dependence of internal replay on the pre-trained convolutional layers, it remains to be confirmed to what extent this component of our model will be useful for other input modalities.

3) The overall approach reminds me of Parisi et al. (2018) in which the authors also 1) use pre-trained convolutional networks as a low-level layer, 2) use network embeddings rather than raw input or pseudo-input for experience replay, 3) do not require task labels. Because of such conceptual analogies, it would be convenient if the authors could explain the potential similarities and differences with this approach. Furthermore, an interesting aspect of this model is the use of temporal context to improve learning. Although the learning of temporally correlated data is probably out of the scope of this study, the authors should speculate whether and how the proposed brain-like model can extend to the temporal domain, a crucial property of biological learning.

We thank the referee for pointing out the Parisi et al. (2018) study, which indeed has several conceptual analogies with our work. In the revised manuscript we discuss the main similarities and differences between their work and ours towards the end of the section “Replay in the brain” in the discussion. Briefly, the main difference is that the method proposed by Parisi et al. does not explicitly learn a generative model to generate the network embeddings to be replayed, but stores them using a recurrent self-organizing network by growing additional neurons for each new experience.

As the referee points out, an appealing property of the method proposed by Parisi et al. is that it is able to take advantage of temporal context and naturally occurring temporal correlations in the input sequences to improve learning. This is an important property of biological learning and indeed an aspect that is missing from our proposed method. In the revised manuscript this is discussed towards the end of the section “Replay in the brain” in the discussion, and we speculate whether aspects of the Parisi et al. approach could be integrated into our method.

4) Of minor concern regards the lack of clarification of some used terminology. For instance, the authors occasionally refer to “true lifelong learning”. Although in the ML/DL literature “continual” and “lifelong” are used interchangeably, these two terms have different meanings in other fields such as robotics and behavioral psychology. The authors should clarify what they mean by true lifelong learning.

We thank the referee for pointing this out. We used “true lifelong learning” in our initial manuscript to refer to a scaled-up version of continual learning (i.e., problems with very many tasks to be learned incrementally). We appreciate that this term was not well chosen and could cause confusing. In the revised manuscript we have now avoided the term “lifelong learning” and we replaced all previous uses by more explicit descriptions of what we mean.

Carr et al. (2011). Hippocampal replay in the awake state: a potential substrate for memory consolidation and retrieval. Nat. Neurosci. 14, 147–153.

Gelbard-Sagiv et al. (2008). Internally generated reactivation of single neurons in human hippocampus during free recall. Science 322, 96–101.

Parisi et al. (2018) Lifelong Learning of Spatiotemporal Representations With Dual-Memory Recurrent Self-Organization. Front. Neurobot. 12:78.

Referee #3:

In the manuscript entitled "Brain-like replay for continual learning with artificial neural networks," the authors tackle the problem of catastrophic forgetting in the task-incremental continual learning setting in which a classifier network has a single readout head, which has to learn an increasing number of classes. To that end, the authors propose a continual learning framework that combines context-dependent gating (XdG), VAE-based replay mechanisms, internal replay, and additional parameter-based regularization techniques such as EWC and SI. The authors discuss these modifications in the context of biological plausibility and demonstrate competitive performance on a range of classification tasks.

The manuscript tackles the exciting problem of continual learning in deep neural networks. It is well written with neat figures and correctly cites the relevant literature. Overall, the results seem appealing for a broader community with a primary interest in deep learning models. The claims about "brain-like" mechanisms, however, seem a little bold, too bold perhaps, and could use better support or else could be moderated.

Main points

Statements about brain-like replay

The manuscript makes bold claims about the involved mechanisms being more brain-like, by incorporating a range of mechanisms pertaining to memory replay and context in a supervised deep learning paradigm. However, at the core, the models studied are deep learning models, and there is no explicit connection or justification for why the present proposal is more brain-like. The current presentation seems misleading in this regard, which risks obscuring, perhaps too much, the underlying premise of this work. To alleviate this shortcoming, one would wish for a more rigorous discussion of experimental predictions of the proposed replay mechanism or offer a comparison to experimental data. Overall, it would seem wise to moderate the statements about the "brain-like replay" mechanism. This cut is not to say that the present work is not exciting or brain-inspired.

We thank the referee for the insightful review. We accept the referee's concern that our statements about "brain-like" replay might have been too bold and that there was insufficient support to justify them. In the revised manuscript we have moderated these statements; we now only describe our proposed modifications and the resulting method as "brain-inspired".

In the revised manuscript we have also extended our discussion of the relation between our proposed replay-based method and replay in the brain. In particular, we now also highlight that there are important aspects of replay in the brain that are missing from our method.

Contribution of different components to the overall performance

The present model makes several extensions and modifications to published replay mechanisms and illustrates their overall positive net effect on continual learning in the incremental class setting. To avoid the impression of a "bag of hacks", what is presently missing is a fine-grained analysis of which model component contributes what to the overall improvements.

To provide insight into the separate and combined effects of our modifications to the standard generative replay approach, in the revised manuscript we have included a series of addition- and ablation-experiments (Fig. 8; see also our response to the final point of referee #1). Besides indicating that internal replay appeared to be our most influential modification (see below for more on that), these new experiments highlight that our various modifications are complementary to each other as their combined effect tended to be greater than the sum of their individual contributions. Moreover, these experiments show that for both permuted

MNIST and class-incremental learning on CIFAR-100, none of our individual modifications were sufficient for the final performance, while all of them except replay-through-feedback were necessary. The contribution of replay-through-feedback is rather to increase efficiency (i.e., no need for two separate models) without substantially hurting performance, although ablating replay-through-feedback sometimes slightly reduced performance as well.

For instance, it would be absolutely desirable to see a quantification of the VAE generator performance using one or several established metrics from the VAE literature to measure both sample quality and diversity instead of merely stating the performance of the overall algorithm. The samples shown in Figure 6d do indeed look "too low", as stated in the manuscript, but it should be quantified. Ideally, such quantification would distinguish between the contributions of the multi-modal Gaussian mixture prior, the context-dependent gating in the feedback connections, and both together to quantify how the presumably improved sample quality improves continual learning performance.

Similarly, the difference between internal replay and vanilla (external?) replay should be quantified. Is this a trick that improves performance? Is this done because it seems more biologically plausible, and it doesn't hurt performance? These questions do not seem addressed in the present version.

In the revised manuscript we have quantified the performance of the VAE generator with and without our various modifications using five different measures. Firstly, we used two traditional measures for evaluating VAE models: average log-likelihood (estimated using importance sampling and a Gaussian observation model) and reconstruction error. These measures suggest that internal replay accounts for a large improvement in VAE performance (see Supplementary Fig. 4), but it is unclear how fair this comparison is given that different input distributions are modelled with internal replay and with replay at the pixel level.

To ensure a fair comparison between samples generated at the internal level versus at the pixel level it is important to first transform them to a common embedding space. We therefore next used the measures Inception Score and Fréchet Inception Distance. The first step of the original versions of these measures is to embed samples using Inception Net, but because our generated internal representations could not be fed into Inception Net we replaced it with a different neural network classifier that used the same pre-trained convolutional layers as in the VAE and that was trained offline on the full CIFAR-100 dataset. This substitution means that our reported scores are not directly comparable with those reported in the literature, but they are valid for the comparisons within our paper (see also Li et al., 2017 *NeurIPS*, <https://arxiv.org/abs/1709.01215>). These measures confirmed that the generated samples replayed by our brain-inspired replay method were substantially better than those replayed by standard generative replay, and that this improvement was for a large part due to internal replay but that the other modifications played a role as well (see Fig. 8a,b).

Finally, because Inception Score and Fréchet Inception Distance do not differentiate between the quality and the diversity of the generated samples, we also reported a modified version (again using our own neural network instead of Inception Net to embed samples) of the recently proposed Precision & Recall curves (Sajjadi et al., 2018 *NeurIPS*, <https://arxiv.org/abs/1806.00035>). These curves showed that both sample quality and sample diversity were substantially improved by our modifications (see Fig. 8c).

Minor suggestions

It isn't entirely clear how XdG context modulation in the generator pass, but not the forward pass, fits with the sought after biological plausibility.

We did not intend to claim that context modulation only in the generator pass is more brain-inspired or biologically plausible than having context modulation also in the forward pass. The brain-inspired part about our “gating based on internal context”-modification is the use of context-dependent processing (in the same sense that XdG was brain-inspired); but the reason that we use context-dependent gating only in the generator pass is “out of necessity” because with class-incremental learning the required context information is not available during the forward pass at test time. In the revised manuscript we adapted the paragraph about gating based on internal context to make this clearer. In the revised manuscript we also removed the paragraph about context-dependent processing in the discussion; this was done partly because this paragraph might have contributed to the confusion and partly to make room for the extra content added during the revision.

Presently the manuscript dedicates a large amount of “real estate” on the justification that replay might be crucially important. While this is an important point, it puts the main findings of the proposed model relatively late in the manuscript Fig. 5-7. To captivate a broader audience, it could be a good idea to shorten the motivation section and put a stronger emphasis on the actual solution and its components (also by dissecting it; see major point above).

In the revised manuscript we have put a stronger emphasis on brain-inspired replay and its ability to incrementally learn new classes from natural images by including two additional analyses dissecting the performance of brain-inspired replay (Figs. 8 & 9) and by shortening parts of the preceding sections.

Figure 7: One could consider adding arrows for the generator labels from the Gaussian mixture model.

In the revised manuscript we have updated this figure, although we decided not to add these arrows as in our opinion they did not make the schematic more interpretable.

REVIEWERS' COMMENTS:

Reviewer #1 (Remarks to the Author):

The authors have proposed a method for continual learning which uses several brain-inspired features to reduce catastrophic forgetting. The method is evaluated in several supervised learning settings, with the most challenging being class-incremental learning on CIFAR-100. The modifications are based on generative replay methods. The first 'brain-inspired' modification is to use generative replay with soft targets rather than hard targets. The other 4 modifications are 1) replay-through-feedback, 2) turning the latent variables into a Gaussian mixture so that samples from different classes can be conditionally generated, 3) gating the generator based on the class of the generated sample, and 4) using pretrained and frozen convolutional layers and only replaying generated hidden states rather than generated inputs. None of these modifications is novel, as all are derived from other published approaches, which would usually detract substantially from the merit of the submission. However, in this case the 4 (or 5, if you count the soft targets) modifications do seem to result in a substantially better outcome: a model that is able to learn 100 classes of CIFAR sequentially, without catastrophic forgetting. Moreover, the modifications hang together well; they are complementary and seamless rather than complex or redundant. While each modification may have value independently, it is the combination which seems to be most effective to retain performance over very long task sequences. The evaluation of the method is thorough, although it only focuses on supervised learning tasks, rather than considering reinforcement learning domains. The ablations are thorough and show that the overall increase of the sum of the modifications is greater than the sum of each individually. The discussion and narrative in the paper is quite insightful and valuable, illuminating both the fundamental problem and the possible solutions.

I recommend acceptance of the manuscript, but there are a few suggestions for the authors to continue improving their paper. First, the discussion is quite lengthy and the language could be tightened up substantially to deliver clearer final points. Second, there are a few typos or awkward wording which could be improved by a final proofreading.

Reviewer #2 (Remarks to the Author):

I consider that the authors have addressed the reviewers' concerns and relevant suggestions.

Reviewer #3 (Remarks to the Author):

Thanks for addressing my questions and concerns. The overall quality of the manuscript has improved and I do not have any further concerns, only some minor comments and cosmetic suggestions:

I understand the problematic of computing the inception score for the internal representations. But, since the current solution does not seem to use the Inception Net, I would suggest also clearly stating that by calling it differently, i.e., not "inception score".

Moreover, it would be great to mention the maximum value of this score. If trained on CIFAR100, I am assuming the maximum value should be 100 (?) whereas the inception score is typically capped at a larger number. I suggest a similar treatment for the Frechet Distance.

Fig. 8: It would be nice to add the respective chance levels in the plots.

P8, first paragraph: "by better quality o r by better diversity" → "or"

Point-by-point response to referees' comments

We thank all referees for their time and effort reviewing our manuscript and for their insightful comments during the revision process. We have incorporated the final suggestions of the referees into the manuscript by shortening the discussion, by renaming two of our measures and indicating bounds for them, and by including chance levels in one of our figures. Below we show the referees' comments (in *italics*), interleaved with our responses.

Referee #1:

The authors have proposed a method for continual learning which uses several brain-inspired features to reduce catastrophic forgetting. The method is evaluated in several supervised learning settings, with the most challenging being class-incremental learning on CIFAR-100. The modifications are based on generative replay methods. The first 'brain-inspired' modification is to use generative replay with soft targets rather than hard targets. The other 4 modifications are 1) replay-through-feedback, 2) turning the latent variables into a Gaussian mixture so that samples from different classes can be conditionally generated, 3) gating the generator based on the class of the generated sample, and 4) using pretrained and frozen convolutional layers and only replaying generated hidden states rather than generated inputs. None of these modifications is novel, as all are derived from other published approaches, which would usually detract substantially from the merit of the submission. However, in this case the 4 (or 5, if you count the soft targets) modifications do seem to result in a substantially better outcome: a model that is able to learn 100 classes of CIFAR sequentially, without catastrophic forgetting. Moreover, the modifications hang together well; they are complementary and seamless rather than complex or redundant. While each modification may have value independently, it is the combination which seems to be most effective to retain performance over very long task sequences. The evaluation of the method is thorough, although it only focuses on supervised learning tasks, rather than considering reinforcement learning domains. The ablations are thorough and show that the overall increase of the sum of the modifications is greater than the sum of each individually. The discussion and narrative in the paper is quite insightful and valuable, illuminating both the fundamental problem and the possible solutions.

I recommend acceptance of the manuscript, but there are a few suggestions for the authors to continue improving their paper. First, the discussion is quite lengthy and the language could be tightened up substantially to deliver clearer final points. Second, there are a few typos or awkward wording which could be improved by a final proofreading.

We thank the referee for their endorsement of our manuscript. In the final manuscript, we have substantially tightened up the discussion section, corrected several typos and replaced some awkward phrases.

Referee #2:

I consider that the authors have addressed the reviewers' concerns and relevant suggestions.

We thank the referee for their previous comments and suggestions, and for accepting our response.

Referee #3:

Thanks for addressing my questions and concerns. The overall quality of the manuscript has improved and I do not have any further concerns, only some minor comments and cosmetic suggestions:

I understand the problematic of computing the inception score for the internal representations. But, since the current solution does not seem to use the Inception Net, I would suggest also clearly stating that by calling it differently, i.e., not “inception score”.

We thank the referee for their positive comments and their continued insightful suggestions. It is indeed the case that our version of the Inception Score does not use the Inception Net. To better reflect this, we now consistently use the term “Modified IS” for this measure and we have made sure that nowhere in the manuscript we refer to this measure simply as “Inception Score” anymore. Similarly, we now consistently refer to our version of the Fréchet Inception Distance as “Modified FID”.

Moreover, it would be great to mention the maximum value of this score. If trained on CIFAR100, I am assuming the maximum value should be 100 (?) whereas the inception score is typically capped at a larger number. I suggest a similar treatment for the Frechet Distance.

In the final manuscript, we have indicated bounds for the Modified IS measure. This measure is indeed bounded from above by 100, which follows from the analysis in the Appendix of Barratt & Sharma (2018, arXiv; <https://arxiv.org/abs/1801.01973>). We have also indicated bounds for the Modified FID measure.

Fig. 8: It would be nice to add the respective chance levels in the plots.

We thank the referee for this suggestion. We have added lines indicating chance levels to this figure. In addition, we have added lines showing the average performance when the base neural network is trained only on the final task or episode, as this can be interpreted as chance performance on all but the last seen task or episode.

P8, first paragraph: “by better quality o r by better diversity” → “or”

Thanks. This typo has been corrected.

Finally, we would like to mention that in the end we decided to follow the suggestion of this referee from the initial round of review to add an arrow for the generator labels to Figure 7c. Thanks for this suggestion and our apologies for not acting on it earlier.